# Predictive accuracy of changes in the inferior vena cava diameter for predicting fluid responsiveness in patients with sepsis: A systematic review and meta-analysis

**Hao Zhang**[1,2,3‡], **Jingyuan Jiang**[1,2,3‡], **Min Dai**[1,2,3], **Yan Liang**[1,2,3], **Ningxiang Li**[1,2,3], **Yongli Gao**[1,2,3]*

1 Department of Emergency Medicine, West China Hospital, Sichuan University/West China School of Nursing, Sichuan University, Chengdu, Sichuan, China, 2 Institute of Disaster Medicine, Sichuan University, Chengdu, Sichuan, China, 3 Nursing Key Laboratory of Sichuan Province, Chengdu, Sichuan, China

‡ HZ and JJ also contributed equally to this work.
* gylzxy1993@163.com(YLG)

## Abstract

### Background

Existing guidelines emphasize the importance of initial fluid resuscitation therapy in sepsis management. However, in previous meta-analyses, there have been inconsistencies in differentiating between spontaneously breathing and mechanically ventilated septic patients.

### Objective

To consolidate the literature on the predictive accuracy of changes in the inferior vena cava diameter (ΔIVC) for fluid responsiveness in septic patients.

### Methods

The Embase, Web of Science, Cochrane Library, MEDLINE, PubMed, Wanfang, China National Knowledge Infrastructure (CNKI), Chinese Biomedical (CBM) and VIP (Weipu) databases were comprehensively searched. Statistical analyses were performed with Stata 15.0 software and Meta-DiSc 1.4.

### Results

Twenty-one research studies were deemed suitable for inclusion. The sensitivity and specificity of ΔIVC were 0.84 (95% CI 0.76, 0.90) and 0.87 (95% CI 0.80, 0.91), respectively. With respect to the distensibility of the inferior vena cava (dIVC), the sensitivity was 0.79 (95% CI 0.68, 0.86), and the specificity was 0.82 (95% CI 0.73, 0.89). For collapsibility of the inferior vena cava (cIVC), the

**Data availability statement:** All data are in the manuscript and/or supporting information files.

**Funding:** The author(s) received no specific funding for this work.

**Competing interests:** The authors have declared that no competing interests exist.

sensitivity and specificity values were 0.92 (95% CI 0.83, 0.96) and 0.93 (95% CI 0.86, 0.97), respectively.

## Conclusion

The results indicated that ΔIVC is as a dependable marker for fluid responsiveness in sepsis patients. dIVC and cIVC also exhibited high levels of accuracy in predicting fluid responsiveness in septic patients.

## Introduction

A 2017 study published in *The Lancet* reported that there were over 48 million cases of sepsis worldwide [1], with a 90-day mortality rate of 35.5% among septic ICU patients [2], resulting in an estimated 5.3 million deaths annually [3]. Sepsis and septic shock, which cause severe acute impairment and deterioration of survivors' chronic health status, have emerged as significant global public health concerns [4,5]. Existing guidelines [6–8] emphasize the importance of initial fluid resuscitation therapy in sepsis management, as this treatment increases cardiac output, enhances organ and tissue perfusion, and reduces mortality [9]. Previous studies have shown that, either alone or in combination, static assessment indicators of fluid responsiveness are not sufficient to accurately and dynamically assess the fluid responsiveness of fluid resuscitation in septic patients, although they can be the most intuitive clinical judgment through physical examination[10,11].The existing guidelines recommend evaluating the patient's circulatory status via dynamic indicators during the initial fluid resuscitation procedure to select additional treatment choices [6–8]. However, determining the optimal volume and target endpoints for fluid resuscitation remains challenging for frontline intensivists in clinical practice [12]. Excessive fluid replacement can lead to serious complications, including tissue edema, hypoxia, and cardiac failure, which can exacerbate clinical outcomes and potentially increase mortality [13,14]. Thus, accurate, objective, and effective dynamic index monitoring methods are essential for evaluating fluid responsiveness in septic patients.

Point-of-care ultrasound (POCUS) is essential for managing sepsis volume because of its convenient and noninvasive monitoring capabilities in real time. Numerous problems, including bleeding and infection, may arise from the invasive procedures and specialized monitoring equipment needed to quantify dynamic parameters such as the stroke volume variation (SVV), volume variability index (PVI), and pulse pressure variation (PPV) [15,16]. This technology helps minimize the dangers of invasive procedures and specialized monitoring equipment. The inferior vena cava (IVC), which is the largest vein in the human body, is linked to right atrial pressure and blood volume. Consequently, the IVC serves as a crucial focal point during ultrasound evaluations of hemodynamics [17–18]. The IVC index obtained through ultrasound serves as a measure of the efficiency of systemic venous return to the heart. The primary purpose of this index to assess the correlation between venous return volume and cardiac function.

Multiple studies have investigated the prognostic importance of changes in the IVC diameter (ΔIVC) for fluid reactivity in hospitalized patients in critical condition. Long et al. [19] found that (ΔIVC) had limited predictive ability for fluid reactivity among critically ill patients who were breathing spontaneously. According to the study conducted by Kim et al. [20], alterations in the diameter of the IVC provide reliable diagnostic precision for forecasting liquid reactivity in spontaneously breathing critically ill individuals. Si et al. [21] proposed that ΔIVC had improved diagnostic accuracy among mechanically ventilated patients when the tidal volume (TV) was ≥ 8 ml/kg compared to a TV ≤ 8 ml/kg. Alvarado et al. [22] reported that ΔIVC had good predictive performance for fluid responsiveness in critically ill adult patients on mechanical ventilation with a TV ≤ 8 ml/kg and without dyspnea or arrhythmia. However, there are still inconsistencies in terms of differentiating between spontaneously breathing and mechanically ventilated septic patients. For mechanically ventilated patients, ΔIVC was referred to as inferior vena cava distensibility (dIVC), and for spontaneously breathing patients, it was referred to as inferior vena cava collapsibility (cIVC). This study aimed to consolidate the literature on the predictive accuracy of changes in IVC diameter for liquid reactivity among septic individuals, thus highlighting the importance of the IVC in managing volume in sepsis patients.

## Materials and methods

This systematic review and meta-analysis carried out was conducted in accordance with the Preferred Reporting Items for Systematic Reviews and Meta-Analysis of Diagnostic Test Accuracy (PRISMA-DTA) statement [23]. The study protocol was formally registered in PROSPERO with the registration number CRD42023469308 to maintain transparency and reliability.

### Selection criteria

(1) The inclusion criteria included studies published in Chinese and English that focused on patients with sepsis or septic shock. The research subjects in the original literature should identify whether they are mechanically ventilated or spontaneously breathing patients. Mechanically ventilated patients should collect ventilator parameters.

(2) The diagnostic method involved measuring patients' ΔIVC via ultrasound. ΔIVC was also referred to as dIVC and cIVC. The purpose of this study was to assess the predictive accuracy of ultrasonographic measurements of ΔIVC for fluid responsiveness in patients with sepsis, utilizing measures such as sensitivity, specificity, true positives (TPs), true negatives (TNs), false positives (FPs), and false negatives (FNs).

(3) The exclusion criteria were as follows: expectant females, individuals younger than 18 years of age, duplicate publications, studies lacking data for a 2x2 chart, review articles, conference presentations, case studies, and animal research.

### Search strategy

The Embase, Web of Science, Cochrane Library, MEDLINE, PubMed, Wanfang, CNKI, CBM and VIP databases were comprehensively searched from inception to February 29, 2024. The search strategy was as follows: (sepsis) OR (sepsis-associated encephalopathy) OR (systemic inflammatory response syndrome) OR (neonatal sepsis) OR (bloodstream infection) OR (infection bloodstream)) AND (inferior vena cava diameter) OR (IVC) OR (inferior vena cava distensibility index) OR (inferior vena cava collapsibility index) OR (inferior vena cava respiratory variation index) OR (dIVC) OR (cIVC) OR (IVC-RVI) OR (caval index)) AND (fluid administration) OR (fluid resuscitation) OR (fluid responsiveness) OR (fluid reactivity) OR (volume responsiveness)). PubMed database which showed in Fig 1.

### Study selection

During the first round of the study selection process, two independent reviewers examined the titles and abstracts of the citations, extracted the data and performed cross-verification. Two researchers conducted the screening and data

#1 "Sepsis"[Mesh] OR "Sepsis-Associated Encephalopathy"[Mesh] OR "Neonatal Sepsis"[Mesh] OR "Systemic Inflammatory Response Syndrome"[Mesh]
Bloodstream Infection*[Text Word] or Infection Bloodstream [Text Word] or Pyemia*[Text Word] or Pyohemia*[Text Word] or Pyaemia*[Text Word] or Septicemia*[Text Word] or Blood Poisoning*[Text Word] or Severe Sepsis [Text Word] or Bacteremia [Text Word] or Sepsis Syndrome*[Text Word] or Poisoning* Blood [Text Word] or Sepsis Severe [Text Word] or Shock Septic [Text Word] or Syndrome* Sepsis [Text Word] or Inflammatory Response Syndrome Systemic [Text Word]
#2 Fluid administration [Text Word] or Fluid resuscitation [Text Word] or fluid responsiveness [Text Word] or fluid reactivity [Text Word] or volume responsiveness [Text Word]
#3 Ultrasound [Text Word] or echocardiography [Text Word] or Ultrasonography [Text Word] or Cardiac ultrasonography [Text Word] or inferior vena cava diameter variability [Text Word] or inferior vena cava diameter [Text Word] or distensibility of the inferior vena cava [Text Word] or dIVC[Text Word] or collapsibility of the inferior vena cava [Text Word] or cIVC [Text Word]
#4 #1 and #2 and #3

**Fig 1. Search strategy of PubMed database.**

extraction on the basis of the literature, and any disagreements were resolved by discussion with a third researcher. Endnote X9 was utilized to manage the literature. The initial screening involved reviewing the article title, followed by assessing the abstract and full text to determine eligibility, excluding any obvious inconsistencies. The following data were extracted: the first author; the source of publication; the date of publication; the duration of the study; the type of study conducted; demographic information about the study population; sample size; sensitivity; specificity; TP, FP, TN, and FN rates; diagnostic methods for fluid reactions; and diagnostic thresholds.

## Assessment of quality

Two separate researchers evaluated the risk of bias in the included literature. Any disagreements were resolved by discussion or by consulting a third party. The quality of the literature was evaluated using the QUADAS-2 tool [24], which assesses aspects such as bias risk and clinical relevance. This tool consists of three tiers and 14 sections.

## Statistical analysis

We conducted the statistical analysis via Stata 15.0 software and Meta-DiSc 1.4. In studies demonstrating heterogeneity, Meta-DiSc 1.4 was utilized to examine threshold effects. We performed Spearman's correlation analysis and merged effect size indicators in cases without a threshold effect. Meta-analysis was performed using Stata 15.0, which yielded the sensitivity, specificity, summary receiver operating characteristic curve (AUC), positive likelihood ratio (PLR), diagnostic odds ratio (DOR), negative likelihood ratio (NLR), and a forest plot. The evaluation of publication bias was conducted through the

utilization of Deeks' funnel plot. Fagan plots were used to analyze the clinical importance of IVC respiratory variability in predicting liquid reactivity. For the subgroup analysis, we first compared mechanically ventilated and nonmechanically ventilated individuals. We analyzed various ventilator parameters including TV, PEEP, and threshold, which are mechanically ventilated.

## Results

### Search results

A total of 885 papers were initially identified across nine databases. By removing 18 duplicates with the help of the Endnote X9 tool and then excluding 833 more through screening, the full texts of the remaining 34 citations were thoroughly reviewed. Ultimately, 21 research studies were deemed suitable for inclusion, including 14 studies published in English and 7 studies published in Chinese [25–45]. (Fig 2).

### Characteristics of the included studies

A total of 21 articles encompassing 1207 patients with sepsis were reviewed. Of these, 15 articles focused on dIVC, whereas six focused on cIVC. (Tables 1 and 2).

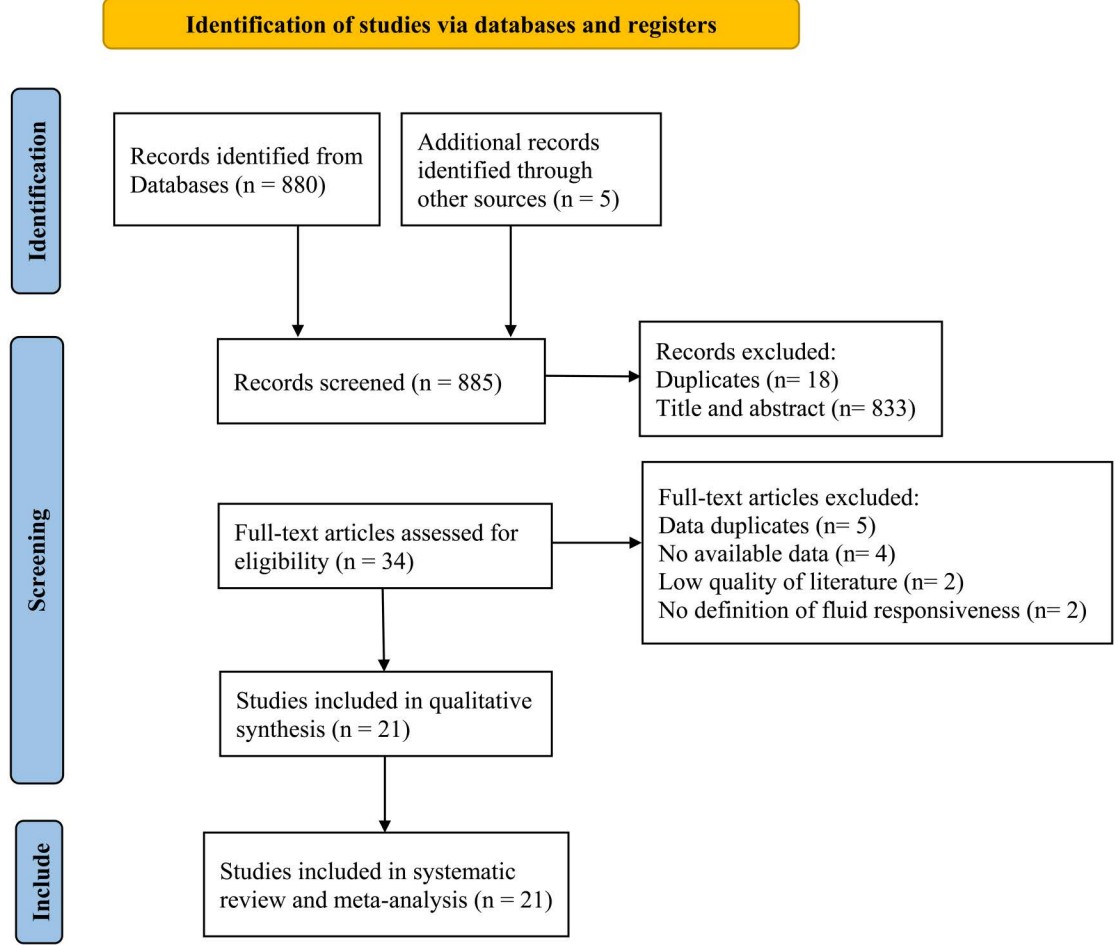

**Fig 2. Flowchart of study selection.**

**Table 1. Main characteristics of the eligible studies.**

| Author | Year | Country | Device | Measure Site | Fluid challenge | Reference Standard | vaso-pressors | MV | MV setting | Index Test | Study period | Threshold | Sample Size |
|---|---|---|---|---|---|---|---|---|---|---|---|---|---|
| Shen | 2019 | China | US (CX50, Holland) | 2 cm from right atrium | 500ml NS/LR over 20min | CO>10% | yes | yes | / | dIVC | 2017.05~2018.10 | 16.5% | 50 |
| Gao | 2021 | China | US (Vivi-id E9,USA) | M-model,2cm the hepatic vein joins the inferior vena cava | 7 ml/kg of LR over 20min | CI ≥15% | yes | yes | VT=6~10ml/kg | dIVC | 2019.01~2020.02 | 17.65% | 27 |
| Yao | 2020 | China | US (Sonosite,USA) | M-model, 2cm from right atrium | 500mL 6% hydroxyethyl starch over 30min | CI > 15% | yes | yes | VT=10ml/kg | dIVC | 2018.10~2019.10 | 28% | 70 |
| Zhang | 2022 | China | US (Resona7,China) | / | / | CI ≥15% | / | yes | / | dIVC | 2020.01~2021.12 | 15.7% | 110 |
| Chen | 2023 | China | US (vivid E9,USA) | / | / | CI>10% | yes | yes | / | dIVC | 2018.01~2020.12 | 11.77% | 98 |
| Zhu | 2016 | China | US (Sonosite,USA) | M-model, 2 cm from right atrium | 500mL of 6% hydroxyethyl starch/NS over 30min | SV ≥15% | yes | yes | / | dIVC | 2013.06~2015.08 | 19.25% | 58 |
| Lu | 2018 | China | / | 2 cm from right atrium | 200ml NS over 10min | CI≥10% | yes | yes | / | dIVC | 2016.01~2017.12 | 20.5% | 65 |
| Orhan | 2022 | Turkey | US (Esaote, Italy) | M-model, 2 cm from right atrium | 10mL/kg of crystal-line solution base-line and 15min | CO ≥15% | no | yes | VT=8ml/kg,PPEP=5 cmH₂O | dIVC | 2018.06.26~2019.07.01 | 17.52 | 40 |
| Charbonn | 2014 | France | US (Suresnes,France) | M-model, Directly above the hepatic vein junction | 7 ml/kg of 6% hydroxyethyl starch 15 min | CI ≥15% | yes | yes | VT=8~10ml/kg | dIVC | / | 21 | 44 |
| Lu | 2017 | China | US (Sonosite, USA) | M-model, 2 cm from right | 200mL NS over 10min | CI≥10% | yes | yes | VT=8~10ml/kg, PEEP=5~12 cmH₂O | dIVC | 2012.01~2015.12 | 20.5 | 49 |
| Theerawit | 2016 | Thailand | US (SonoSite,USA) | M-model, 2 cm from right atrium | 1000 ml NS/500ml 6% hydroxyethyl starch/5% human albumin over 30min | CO ≥15% | yes | yes | VT ≥8ml/kg PEEP=8~10 cmH₂O | dIVC | 2012.11~20.13.12 | 10.2 | 29 |
| Saber | 2022 | Egypt | / | M-model, 2 cm from right atrium | 500mL NS 10min, Two measurements。 | CO ≥15% | yes | yes | VT=6~8ml/kg, PEEP=0~5 cmH₂O | dIVC | 2017.10~2018.10 | 14.5 | 40 |
| Feissel | 2004 | France | / | M-model, 3cm from right atrium | 8ml/kg 6% hydroxyethyl starch 20min | CO ≥15% | / | yes | VT ≥8~10mL/kg | dIVC | / | 12 | 39 |
| Abdelma | 2023 | Egypt | US(GE Health-Care Vivid,USA) | / | 30ml/kg of crystal-line solution | CO ≥15% | yes | yes | VT=8ml/kg,PEEP=5 cmH₂O | dIVC | 2021.09~2022.03 | 13.3 | 34 |
| He | 2023 | China | US(Zonare,USA) | the IVC under the raphe | 200mL NS over 10min | SV ≥15% | / | yes | PEEP=4 cmH₂O | dIVC | 2018.04~2021.02 | 16.5 | 102 |

*(Continued)*

**Table 1.** (Continued)

| Author | Year | Country | Device | Measure Site | Fluid challenge | Reference Standard | vaso-pres-sors | MV | MV setting | Index Test | Study period | Threshold | Sample Size |
|---|---|---|---|---|---|---|---|---|---|---|---|---|---|
| Preau | 2017 | France | US(vivid-i/ Vivid-S5,USA) | 1.5 to 2 cm Hepatic vein junction with IVC/3–4 cm from right atrium | 500ml 4% Gelatin over 30min | SV ≥15% | yes | no | / | cIVC | 2011.11~2014.01 | 48 | 90 |
| Zhao | 2016 | China | | 2 cm Hepatic vein junction with IVC | 500ml 6% hydroxyethyl starch 30min | CI ≥ 15% | yes | no | / | cIVC | 2013.10~2014.03 | 12.9 | 42 |
| Caplan | 2020 | France | US(Vivid-i/ Vivid-S5,USA) | 4 cm from right atrium | 500ml 4% Gelatin over 30min | SV ≥10% | yes/ NE | no | / | cIVC | 2011.11~2015.05 | 44 | 81 |
| Elsaeed | 2022 | Egypt | US(GE,USA) | M model lower edge of the ribcage | 7 ml/kg of LR over 30min | CI ≥ 15% | no | no | / | cIVC | / | 35 | 40 |
| Murat | 2016 | Turkey | US(Philips EPIQ 5,USA) | M model, 0.5 to 3 cm from right atrium–hepatic vein junction | PLR | CI ≥ 15% | / | no | / | cIVC | / | 35 | 44 |
| Perrine | 2018 | France | US(vivid-i/ Vivid-S5,USA) | 1.5 to 2 cm Hepatic vein junction with IVC/3–4 cm from right atrium | 500ml 4% Gelatin over 30min | Velocity time inte-gral≥10% | yes/ NE | no | / | cIVC | 2012.05~2015.05 | 39 | 55 |

Ultrasonographic=US, inferior vena cava=IVC, inferior vena cava diameter=ΔIVC, inferior vena cava distensibility=dIVC, inferior vena cava collapsibility=cIVC, MV=me-chanical ventilation; PEEP=positive end expiratory pressure, tidal volume=VT, NS=Normal saline, PLR=passive leg raising, CO=cardiac output, CI=cardiac index, SV=stroke volume,/=not reported, NE=Norepinephrine.

**Table 2. 2-by-2 table of the eligible studies.**

| Author | Year | TP | FP | FN | TN | Total | Sensitivity | Specificity | AUC |
|--------|------|----|----|----|----|-------|-------------|-------------|-----|
| Shen | 2019 | 20 | 7 | 5 | 18 | 50 | 80% | 72% | 0.777 |
| Gao | 2021 | 19 | 1 | 0 | 7 | 27 | 100.00% | 87.50% | 0.924 |
| Yao | 2020 | 34 | 2 | 5 | 29 | 70 | 87.20% | 93.10% | 0.94 |
| Zhang | 2022 | 44 | 7 | 22 | 37 | 110 | 66.00% | 85.00% | 0.709 |
| Chen | 2023 | 53 | 28 | 3 | 14 | 98 | 95.08% | 34.14% | 0.664 |
| Zhu | 2016 | 17 | 3 | 15 | 23 | 58 | 53.10% | 88.50% | 0.733 |
| Lu | 2018 | 19 | 4 | 12 | 30 | 65 | 60.30% | 89.70% | 0.826 |
| Orhan | 2022 | 18 | 5 | 5 | 12 | 40 | 77.50% | 72.50% | 0.833 |
| Charbonn | 2014 | 10 | 7 | 16 | 11 | 44 | 38% | 61% | 0.43 |
| Lu | 2017 | 18 | 5 | 9 | 17 | 49 | 67% | 77% | 0.805 |
| Theerawit | 2016 | 12 | 3 | 4 | 10 | 29 | 75% | 76.90% | 0.688 |
| Saber | 2022 | 17 | 2 | 3 | 18 | 40 | 85% | 90% | 0.913 |
| Feissel | 2004 | 15 | 2 | 1 | 21 | 39 | 93% | 92% | / |
| Abdelma | 2023 | 21 | 0 | 2 | 11 | 34 | 91.30% | 100% | 0.925 |
| He | 2023 | 35 | 7 | 19 | 41 | 102 | 65.4 | 84.9 | 0.858 |
| Preau | 2017 | 42 | 4 | 8 | 36 | 90 | 84% | 90% | 0.89 |
| Zhao | 2016 | 32 | 0 | 0 | 10 | 42 | 100% | 100% | 0.917 |
| Caplan | 2020 | 38 | 1 | 3 | 39 | 81 | 93% | 98% | 0.98 |
| Elsaeed | 2022 | 23 | 1 | 1 | 15 | 40 | 95.8 | 93.7 | 0.97 |
| Murat | 2016 | 18 | 3 | 5 | 18 | 44 | 78% | 85% | 0.825 |
| Perrine | 2018 | 27 | 3 | 2 | 23 | 55 | 93% | 88% | 0.93 |

true positives = TP, false positives = FP, false negatives = FN, true negatives = TN, area under the receiver operating characteristic curve = AUC.

## Assessment of quality

Using Review Manager 5.3 software, the quality of all included studies was assessed using the QUADAS-2 tool. The findings indicated that the literature was generally high-quality, with potential biases regarding patient selection, the index test, and the reference standard (Figs 3 and 4).

## Statistical analysis

**Threshold effect test.** The threshold effect test was performed using Meta-DiSc 1.4 software. The results revealed Spearman correlation coefficients of -0.440 (P = 0.046) for ΔIVC, -0.157 (P = 0.576) for dIVC, and -0.543 (P = 0.266) for cIVC. These findings suggest a potential threshold effect on ΔIVC, whereas no threshold effect was observed on dIVC and cIVC, allowing their effect sizes to be combined.

**Meta-analysis.** The sensitivity and specificity of ΔIVC were 0.84 (95% CI 0.76, 0.90) and 0.87 (95% CI 0.80, 0.91), respectively. The sensitivity and specificity of dIVC were 0.79 and 0.82, respectively. The sensitivity and specificity of cIVC were 0.92 and 0.93, respectively. The results for the other indicators are presented in detail in Table 3. The forest plots depicting the results of the meta-analysis are provided in the appendix of the manuscript (Figs 5-16).

In Deeks' funnel plots, the values were 0.28, 0.23, and 0.60, indicating that the included studies had no publication bias (Figs 17-19). The 50% predictive probability results were 86%, 82%, and 93%, and the negative posttest probabilities were 16%, 21%, and 8%, respectively (Figs 20-22).

Currently, there is no evidence that a single evaluation parameter or index can be utilized as an endpoint on its own to direct fluid resuscitation in sepsis patients. Our subgroup analysis is based on previous systematic review results,

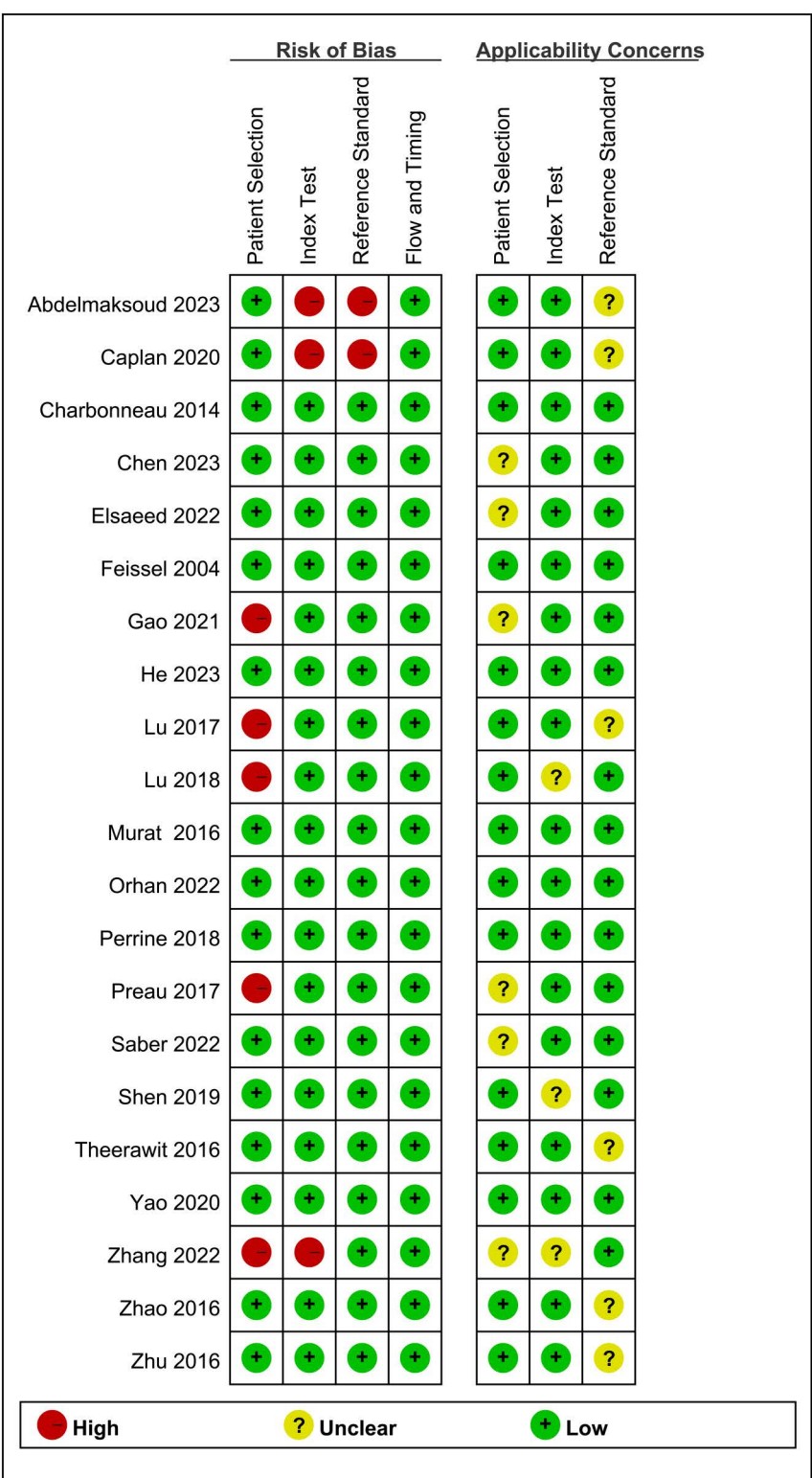

**Fig 3. Quality assessment of diagnostic accuracy studies criteria for the included studies.**

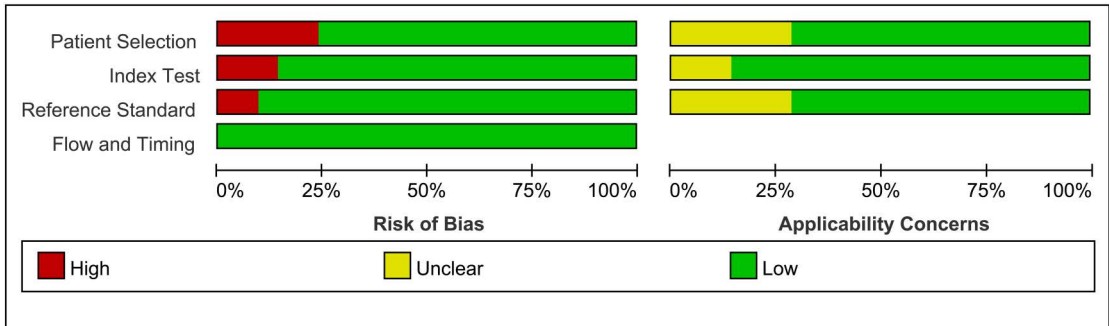

**Fig 4. Quality assessment of diagnostic accuracy studies criteria for the included studies.**

**Table 3. ΔIVC, dIVC and cIVC of meta-analysis.**

| Group | sensitivity | specificity | PLR | NLR | DOR | AUC |
|---|---|---|---|---|---|---|
| ΔIVC | 0.84 | 0.87 | 6.26 | 0.18 | 34.03 | 0.92 |
| | 0.76~0.90 | 0.80~0.91 | 3.93~9.97 | 0.12~0.29 | 15.09~76.76 | 0.89~0.94 |
| dIVC | 0.79 | 0.82 | 4.42 | 0.26 | 17.10 | 0.88 |
| | 0.68~0.86 | 0.73~0.89 | 2.83~6.92 | 0.17~0.40 | 8.10~36.09 | 0.84~0.90 |
| cIVC | 0.92 | 0.93 | 13.52 | 0.09 | 156.62 | 0.97 |
| | 0.83~0.96 | 0.86~0.97 | 6.16~29.66 | 0.04~0.19 | 40.11~611.56 | 0.95~0.98 |

positive likelihood ratio=PLR, negative likelihood ratio=NLR, diagnostic odds ratio=DOR, area under the receiver operating characteristic curve=AUC.

basic ventilator parameters (TV=6-8mL/kg, PEEP=5 cmH2O), and dose selection for fluid resuscitation. Each enrolled patient with septic shock was categorized into the low-volume (200ml), medium-volume (7 mL/kg fluid), or high-volume (above 500 mL) fluid group according to the infusion volume. We conducted subgroup analyses based on TV, PEEP, infusion volume, and diagnostic threshold. A meta-regression analysis was conducted on ΔIVC in the dIVC and cIVC, revealing a statistically significant difference of SE=2.022 (95% CI: 1.81–31.48, P=0.0081), which could account for the observed heterogeneity. Subgroup analyses were then performed based on TV(P=0.580), PEEP(P=0.732), infusion volume(P=0.417), and diagnostic threshold (P=0.472) in the dIVC, as well as infusion volume(P=0.791) and diagnostic threshold (P=0.846) in the cIVC. The analyses did not yield significant results (Tables 4–5).

## Discussion

This study analyzed a total of 21 papers involving 1207 septic patients. This meta-analysis is the first study to assess the predictive value of cIVC and dIVC for liquid reactivity among septic individuals who are mechanically ventilated and spontaneously breathing. The findings indicated that ΔIVC had good overall performance in predicting fluid reactivity in septic patients, with a combined sensitivity, specificity, and AUC of 0.84, 0.87, and 0.92, respectively, exceeding the values reported in previous systematic reviews [19,20,46–48].

The subgroup analysis could not determine the effect of vasopressors on fluid resuscitation because only two of the assessed studies included patients who did not receive vasopressors. Vasopressors and fluid resuscitation are essential components of hemodynamic support for treating patients with septic shock [49]. Vasopressors may also be utilized until fluid resuscitation is finished, according to ESICM recommendations [6]. According to previous research, treating patients with septic shock necessitates multiangle assessment, individualized treatment, quantified and targeted fluid management

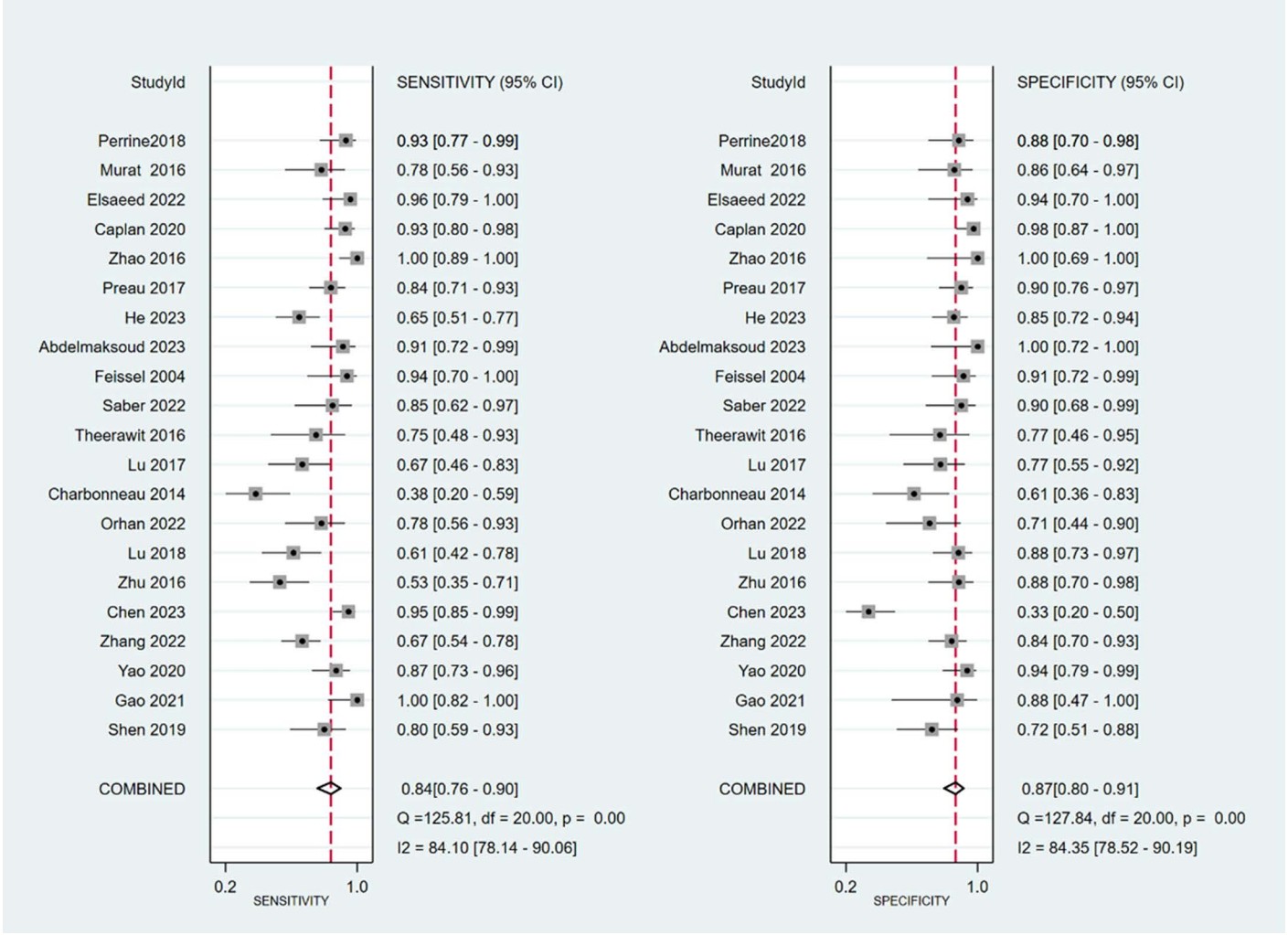

**Fig 5. Forest plot of sensitivity and specificity in the diagnosis of ΔIVC.**

and assessment, and prudent and standardized use of vasopressors, all of which are helpful in the resuscitation of patients experiencing early septic shock [50].

A rational fluid approach necessitates that we view septic shock as a multiphase illness, and numerous indicators or techniques are combined on the basis of the patient's volume responsiveness. Subgroup analyses by Long [19] et al. and Kim et al. [20]. demonstrated that ΔIVC was an effective predictor of fluid reactivity, with AUCs of 0.85 and 0.87, respectively; however, ΔIVC did exhibit lower sensitivity. The cIVC showed excellent performance in terms of predicting fluid reactivity in spontaneously breathing septic individuals, displaying superior diagnostic accuracy compared with the dIVC in mechanically ventilated septic patients. This finding is consistent with the study by Orso et al. [47] but contrary to those of Zhang et al. [46], Long et al. [19] and Kim et al. [20], possibly due to the inclusion of a diverse range of critically ill patients. However, Long et al. [19]. and Orso et al. [47]. included both adults and children in their studies. Since children have different physiological characteristics, such as vascular compliance, respiratory rate, and TV, than adults, caution should be taken when comparing and analyzing the results.

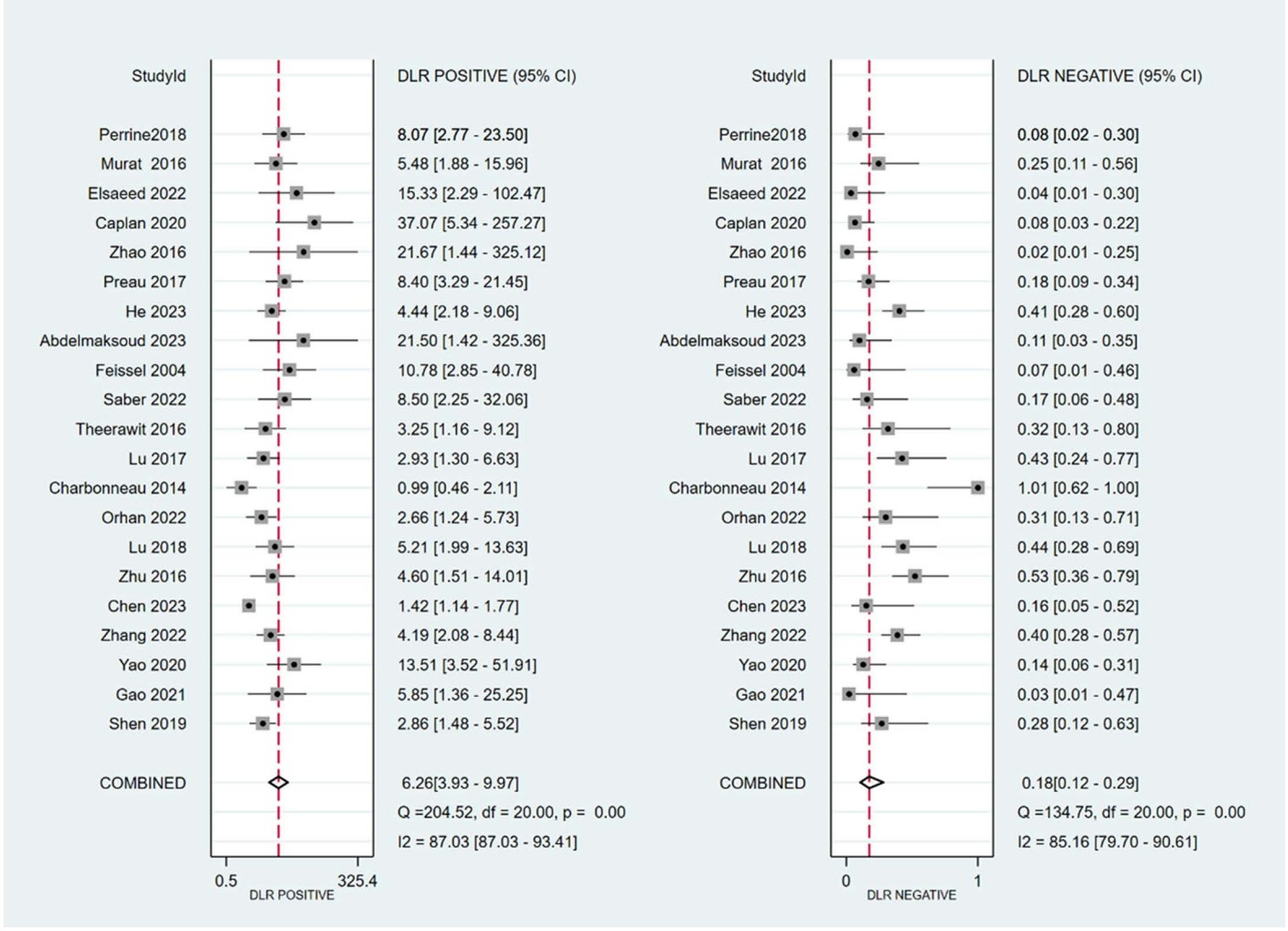

**Fig 6. Forest plot of PLR and NLR in the diagnosis of ΔIVC.**

The latest edition of the Sepsis and Infectious Shock Guidelines, which was published by the Surviving Sepsis Campaign, recommends the use of dynamic measurements to assess fluid responsiveness [6]. Ultrasound is utilized for hemodynamic evaluation with a focus on volume. The IVC, which is the largest venous trunk in the body, is closely linked to right atrial pressure and blood volume. It is pliant and easily dilated, and its internal diameter and degree of collapse are used to evaluate volume status in critically ill patients [51]. ΔIVC provides a comprehensive three-dimensional assessment of the internal diameter of the IVC, aligning with current practices. Meta-regression analysis of ΔIVC among different respiratory tract types revealed significant differences, with dIVC and cIVC potentially contributing to the observed heterogeneity. While ΔIVC was termed dIVC and cIVC in mechanically ventilated and spontaneously breathing patients, respectively, subgroup analyses in related studies revealed differences only in respiratory or disease types, thus lacking a solid scientific basis for volume assessment in septic patients. Studies have suggested that the ΔIVC performs better in

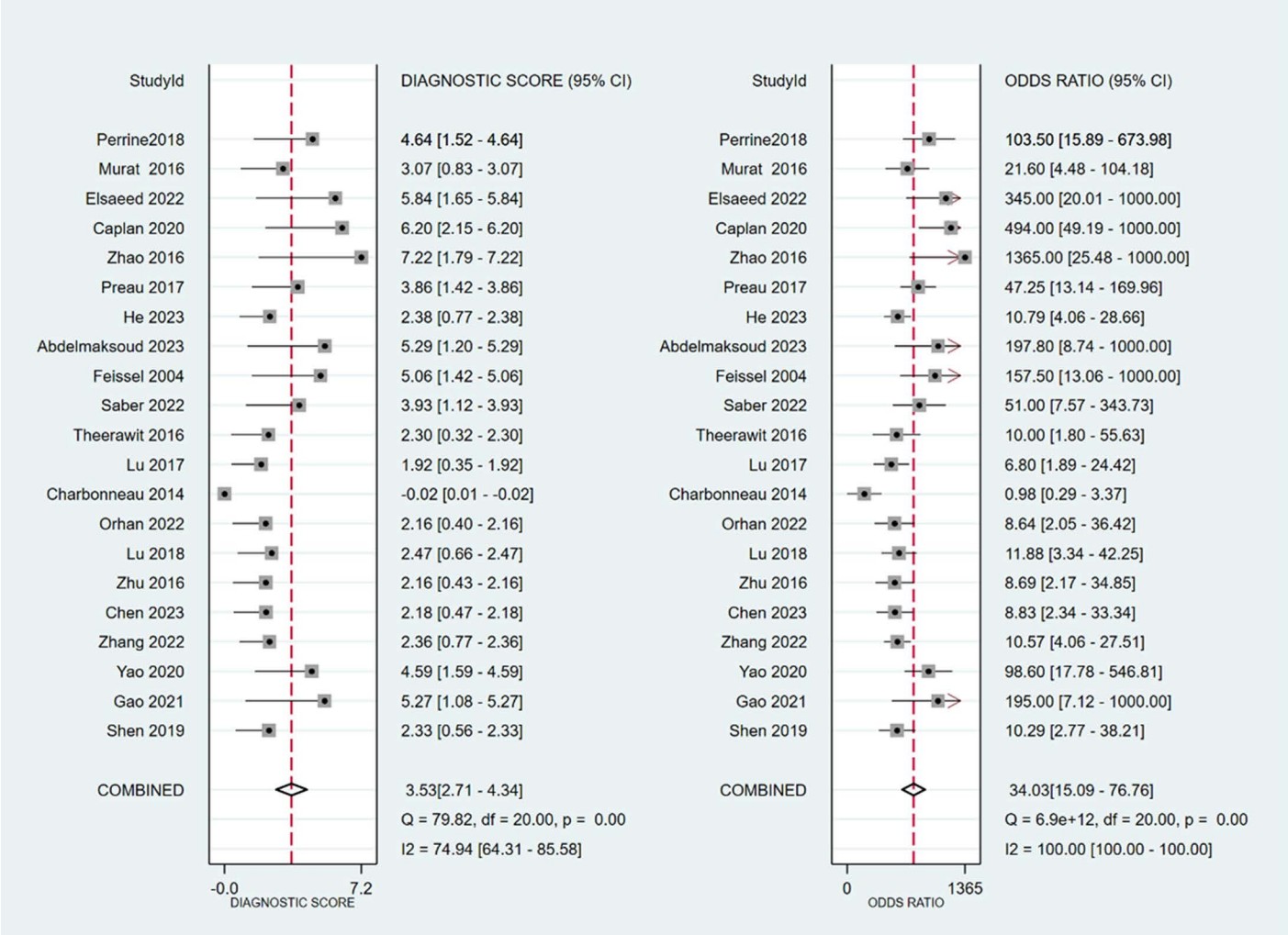

**Fig 7. Forest plot of DOR in the diagnosis of ΔIVC.**

predicting fluid responsiveness in mechanically ventilated patients than in spontaneously breathing patients [19,20,46], possibly because of factors such as hypovolemia or changes in intrathoracic pressure during respiration. dIVC has shown promise in predicting fluid responsiveness in mechanically ventilated septic patients, with higher sensitivity, specificity, and AUC values than those reported in previous studies. This improvement may be attributed to the inclusion of more recent studies in the analysis. This mechanism arises from the varying breathing patterns of patients, leading to distinct trends in the variation in the inner diameter of the inferior vena cava during inspiration [52]. Among individuals who breathe spontaneously, negative pressure is present in the chest at the end of expiration. Fluctuations in intrathoracic pressure can result in a decrease in pressure within the right atrium, thereby increasing the speed of venous return. Conversely, during mechanical ventilation, the intrathoracic pressure increases during the inspiratory phase. This elevation in pressure results in an increase in right atrial pressure, which subsequently obstructs venous return.

The determination of fluid responsiveness via dIVC was more accurate in septic patients with ventilator parameters of TV ≤ 8 ml/kg or PEEP ≤5 cmH$_2$O than in those with TV ≥ 8 ml/kg or PEEP ≥5 cmH$_2$O. However, meta-regression analysis

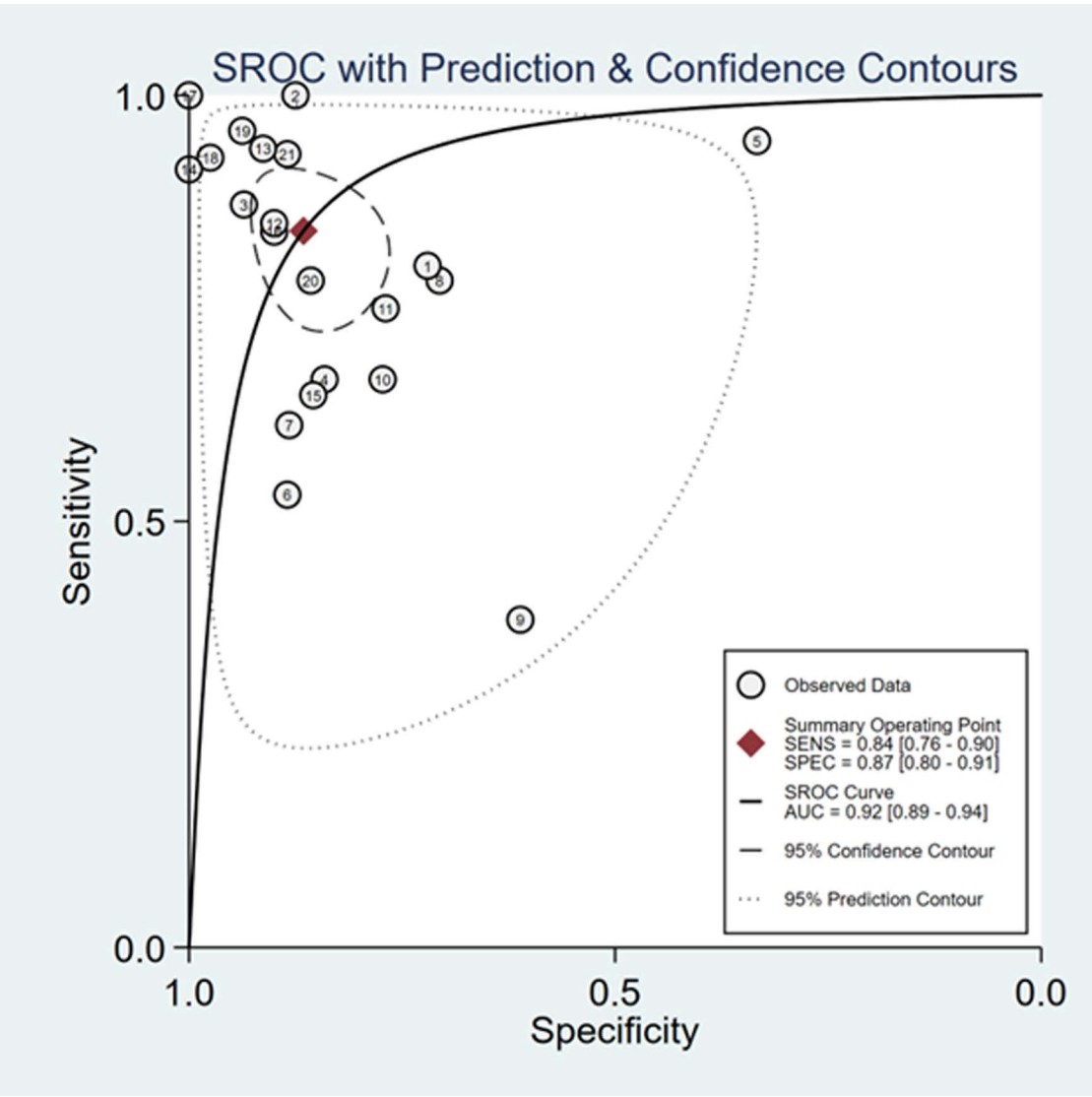

**Fig 8. Summary receiver operator characteristic (SROC) curve in the prediction of ΔIVC. SENS = sensitivity, SPEC = specificity, AUC = area under the receiver operating characteristic curve.**

of TV and PEEP did not reveal any significant moderators, and subgroup analysis did not yield any statistically significant differences. These findings align with a study conducted by Alvarado et al. [22]. The results reported by Si et al. [21] were also consistent with our results with respect to PEEP, but those authors reported opposite results concerning tidal volume. In mechanically ventilated patients, the ventilator controls inspiratory flow, resulting in positive pressure and elevated intrathoracic pressure and leading to changes in the IVC diameter [53]. Studies have shown that the use of low TV ventilation modes in septic patients can improve pulmonary oxygenation and reduce airway pressure, thereby lowering the risk of lung injury. Additionally, larger TV may lead to alveolar overdistension, impair pulmonary blood flow perfusion, and redistribute fluids, ultimately compromising the effectiveness of fluid resuscitation [54]. Research indicates that appropriate

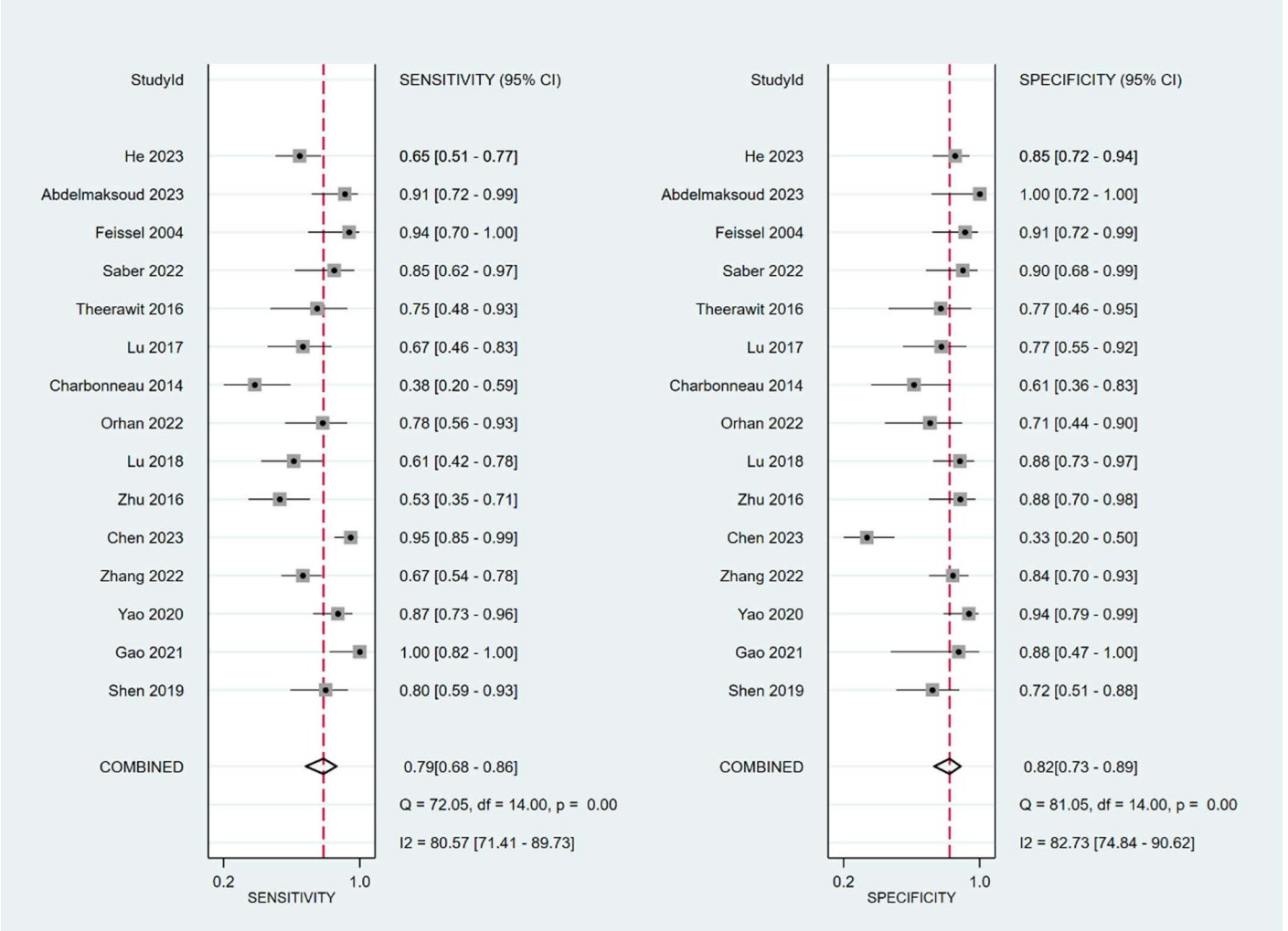

**Fig 9. Forest plot of sensitivity and specificity in the diagnosis of dIVC.**

PEEP can enhance alveolar ventilation, improve oxygenation, and reduce the incidence of pulmonary edema. However, excessively high PEEP may decrease cardiac return, adversely affecting hemodynamics and increasing fluid demands [55]. It is crucial to optimize respiratory conditions through precise ventilator settings to ensure lung protection while avoiding excessive fluid overload. For example, if chest ultrasound demonstrates adequate cardiac preload but PEEP settings result in insufficient circulation, adjustments to fluid administration may be necessary. However, when a patient experiences excessive circulating volume load, the inner diameter of the inferior vena cava expands, and the amplitude of its movement diminishes. Previous studies have several limitations, such as small sample sizes and a lack of differentiation between spontaneously breathing and mechanically ventilated septic patients. A more comprehensive study on ventilator parameter settings is necessary to validate these results.

In septic patients, ΔIVC was found to be more effective than the comparison group in terms of predicting fluid responsiveness with a 500-ml infusion volume and a dIVC threshold >16.5%. However, a dIVC threshold >16.5% had a sensitivity of 0.69, which was lower than that of the comparison group. Subgroup analysis of the cIVC threshold did

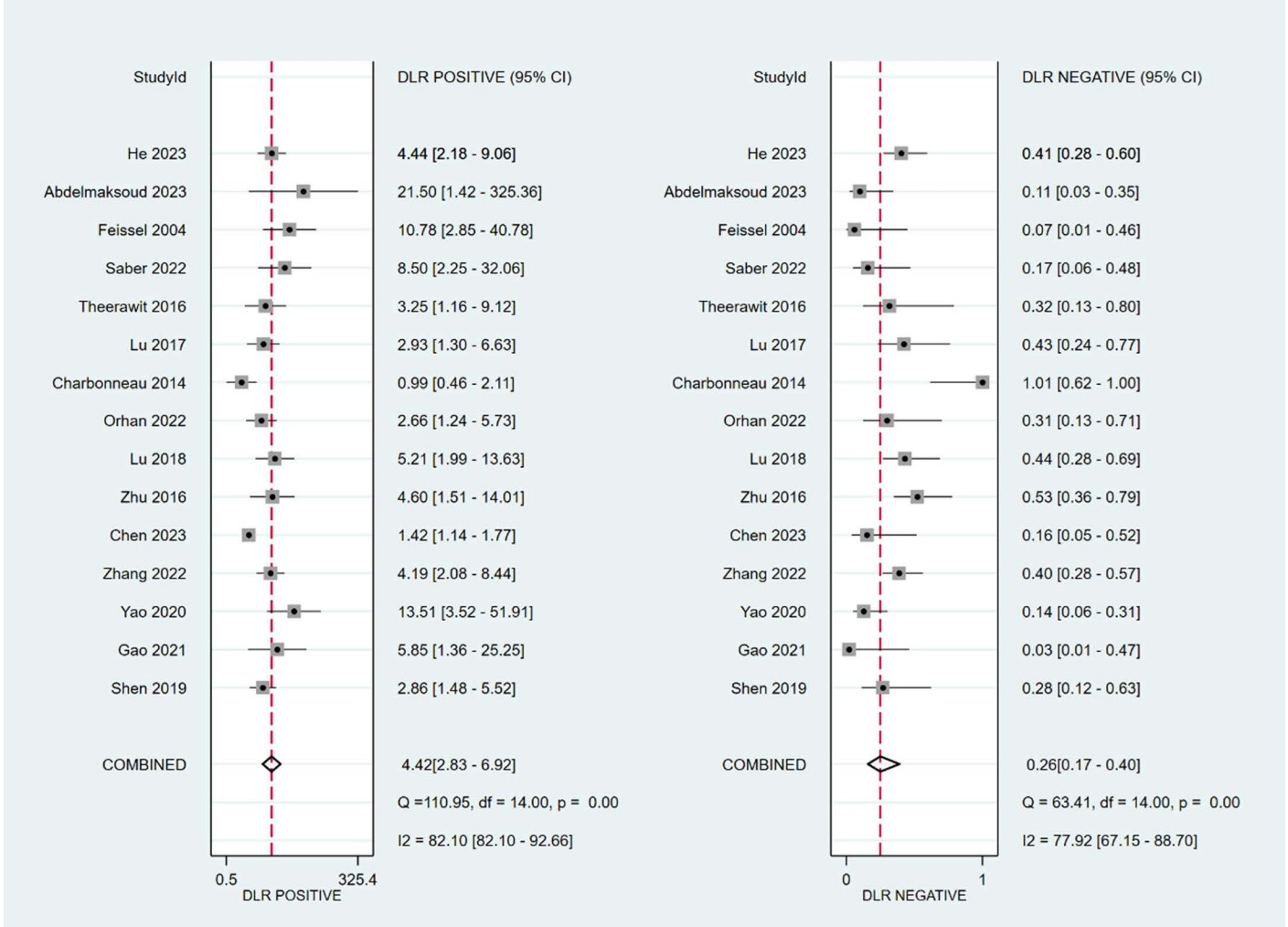

**Fig 10. Forest plot of PLR and NLR in the diagnosis of dIVC.**

not reveal significant differences in diagnostic efficacy. Meta-regression analysis revealed that none of the moderators were significant, and no statistically significant differences were observed. The studies included dIVC and cIVC thresholds ranging from 10.2 to 28 and 12.9 to 48, respectively. Changes in the chest wall and lung compliance may impact pulse pressure changes but do not affect volume status, potentially altering thresholds and accuracy of assessment [56]. Corl et al. [57] studied 124 spontaneously breathing critically ill patients in the ICU by measuring the IVC diameter before and after 500 ml of fluid rehydration and reported that an IVC dilatation index >25% predicted increased cardiac output postfluid infusion in spontaneously breathing patients. The area under the curve was 0.84. Airapetian et al. [58] reported 97% sensitivity and 31% specificity in predicting fluid response in autonomously breathing patients when the ΔIVC was >42%. According to the Frank–Starling curve, only approximately 50% of patients are volume-responsive enough to benefit from aggressive fluid resuscitation [59]. Small-volume liquid infusion inevitably decreases the critical values of C0, CI and other indicators for judging volume responsiveness, which requires researchers to have very

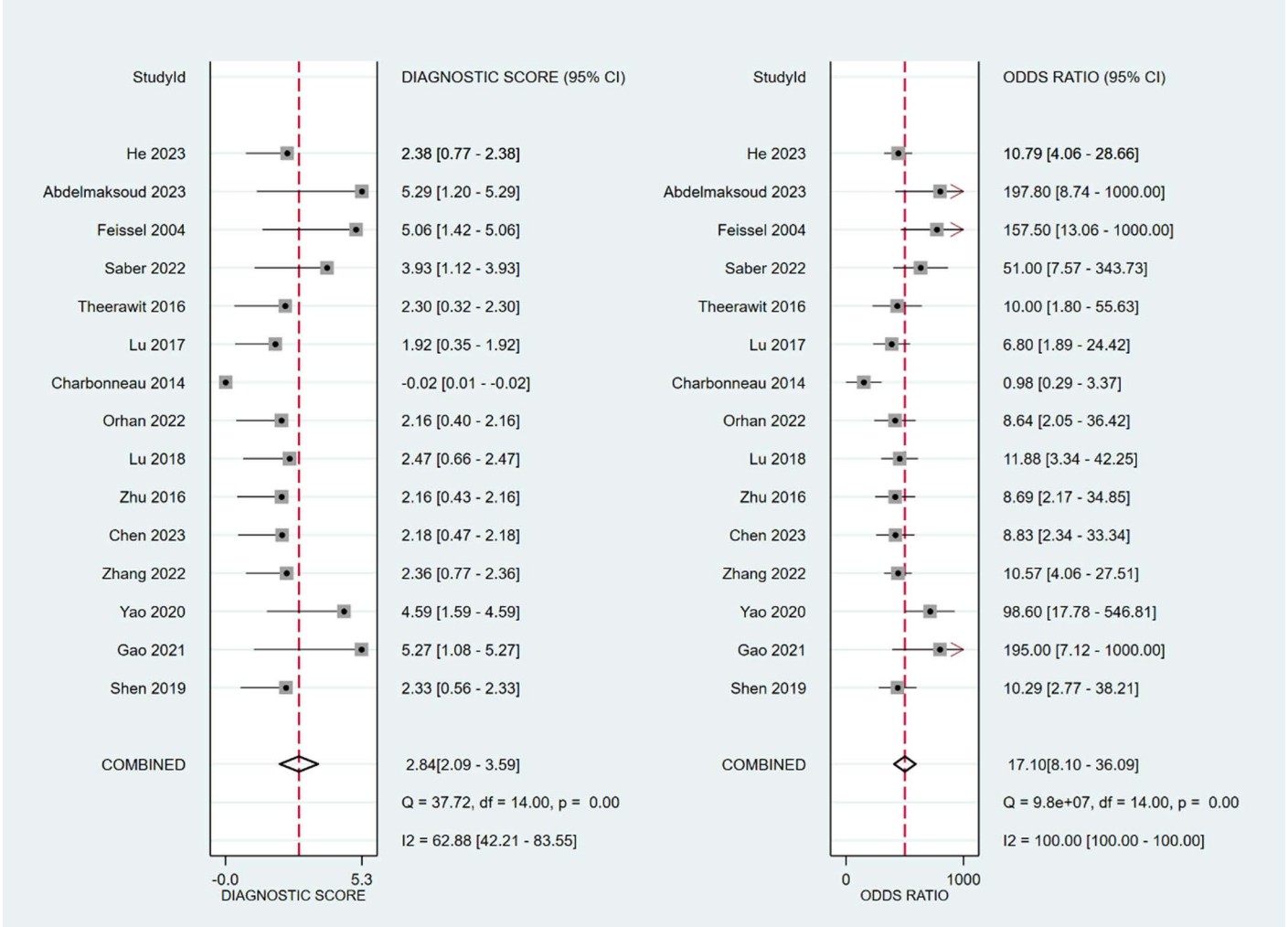

**Fig 11. Forest plot of DOR in the diagnosis of dIVC.**

skilled ultrasound technology to reduce errors caused by operation and subjective judgment. Using threshold-guided fluid resuscitation to guide fluid resuscitation in patients with sepsis can more effectively increase the patient's cardiac output and correct and maintain the intracellular and extracellular electrolyte balance [9]. Patients with septic shock who complete 30 mL/kg fluid resuscitation within 1–2 hours can significantly reduce 28-day mortality while improving organ function [60]. Fluid overload on ICU day 1 may lower in-hospital mortality risk but could increase mortality after day 3. This indicates that early adequate fluid resuscitation is beneficial, whereas strict fluid restriction in later phases is critical to avoid excessive overload [61]. A high fluid balance is associated with increased mortality in severe sepsis and septic shock patients, while low-volume resuscitation within the initial 24 hours markedly reduces mortality. This suggests that lower fluid resuscitation thresholds may be safer for certain patient populations. Lauralyn et al. [62] suggest that critically ill patients exhibit significant heterogeneity, rendering standardized therapeutic measures increasingly controversial and challenging both theoretically and practically. Initial fluid resuscitation should adhere to

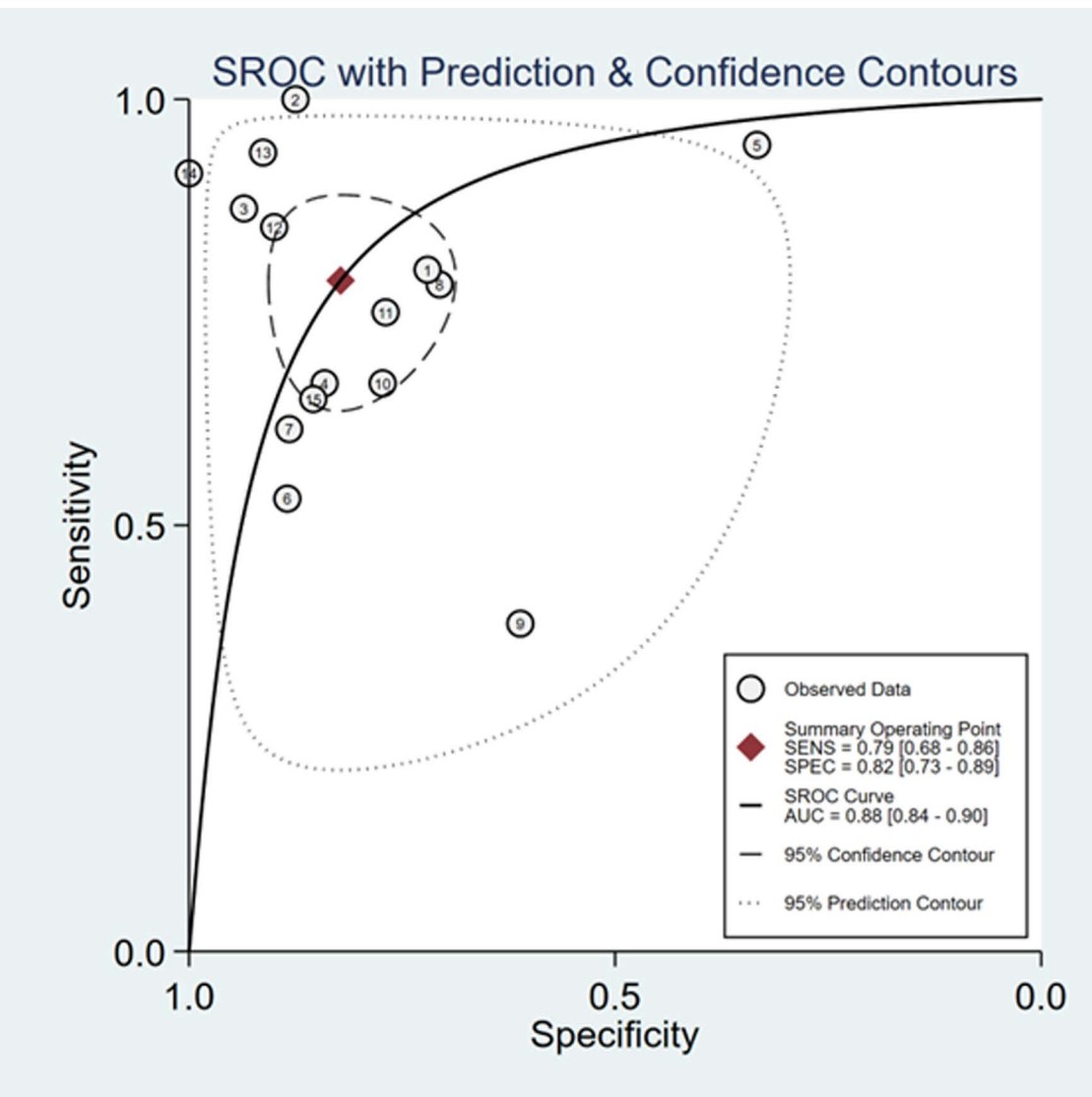

**Fig 12. Summary receiver operator characteristic (SROC) curve in the prediction of dIVC. SENS = sensitivity, SPEC = specificity, AUC = area under the receiver operating characteristic curve.**

an individualized approach. The four-phase resuscitation strategy requires close coordination and timely adjustments based on the patient's clinical status. However, ineffective implementation persists due to clinicians' failure to strictly follow guidelines and insufficient clinical experience.

## Limitations

First, heterogeneity in measurement protocols primarily arises from differences between mechanically ventilated and non-ventilated patient cohorts. Second, critically ill patients demonstrate significant heterogeneity that those with

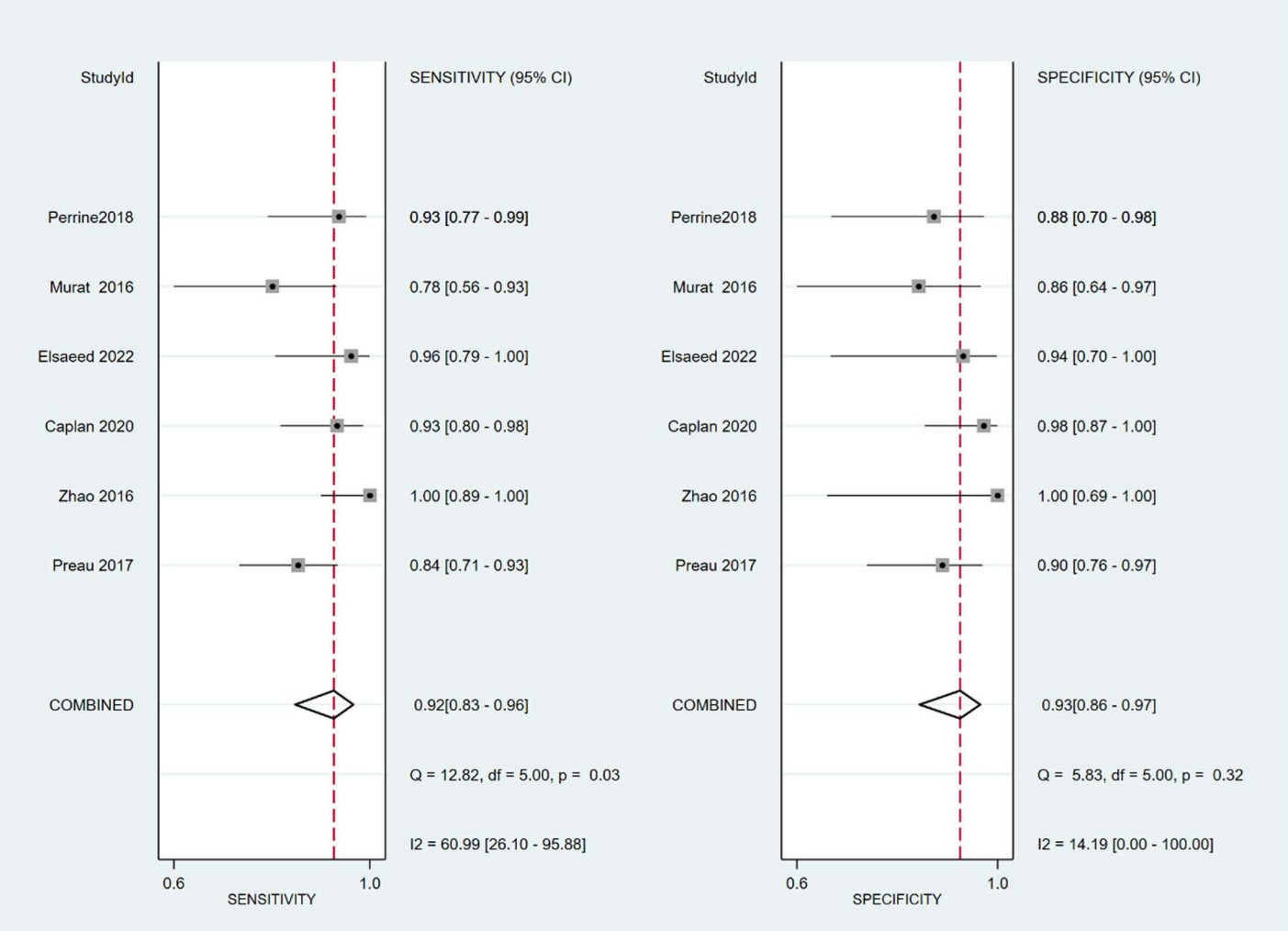

**Fig 13. Forest plot of sensitivity and specificity in the diagnosis of cIVC.**

heart failure, kidney disease, or other comorbidities may exhibit marked variations in fluid resuscitation efficacy. This necessitates the integration of dynamic parameters, such as ultrasound-guided assessments combined with pulse pressure variation (PPV) or fluid challenge tests, when evaluating fluid resuscitation in septic patients. Additionally, blood lactate levels should be combined to comprehensively assess resuscitation outcomes. Furthermore, the impact of concomitant vasopressor use during fluid resuscitation on the trial results must be considered in our study.

## Conclusion

The findings demonstrate that selecting parameters with tidal volume ≤8 mL/kg, PEEP ≤5 cmH$_2$O, and fluid volume ≥500 mL is more effective for accurately assessing fluid resuscitation in mechanically ventilated septic patients. For non-ventilated septic patients, maintaining a fluid volume ≥500 mL remains clinically significant. Overall, ΔIVC showed strong predictive value for liquid reactivity in septic individuals, with the dIVC and cIVC displaying good and excellent accuracy in predicting fluid reactivity in mechanically ventilated and spontaneously breathing septic individuals.

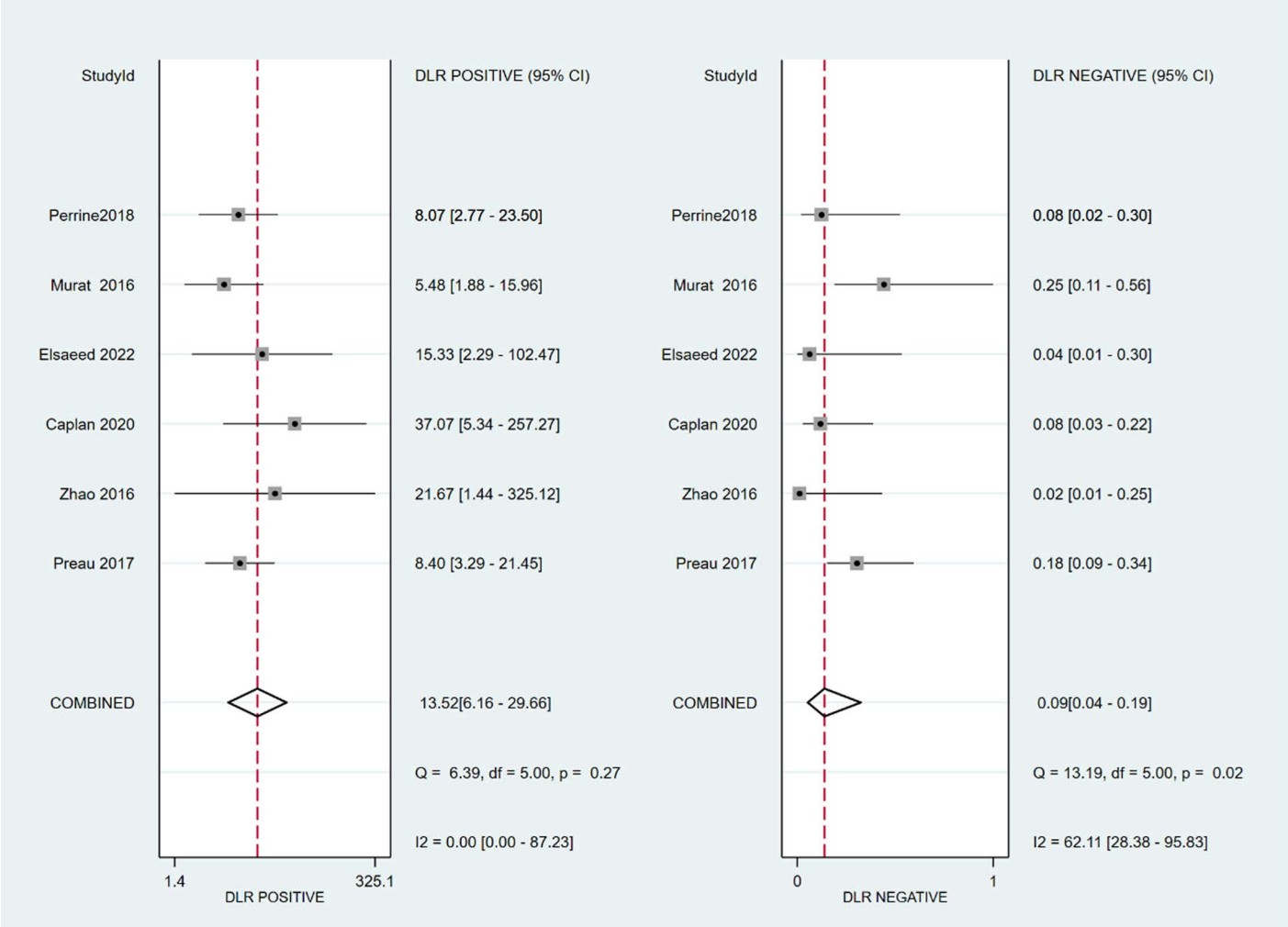

**Fig 14. Forest plot of PLR and NLR in the diagnosis of cIVC.**

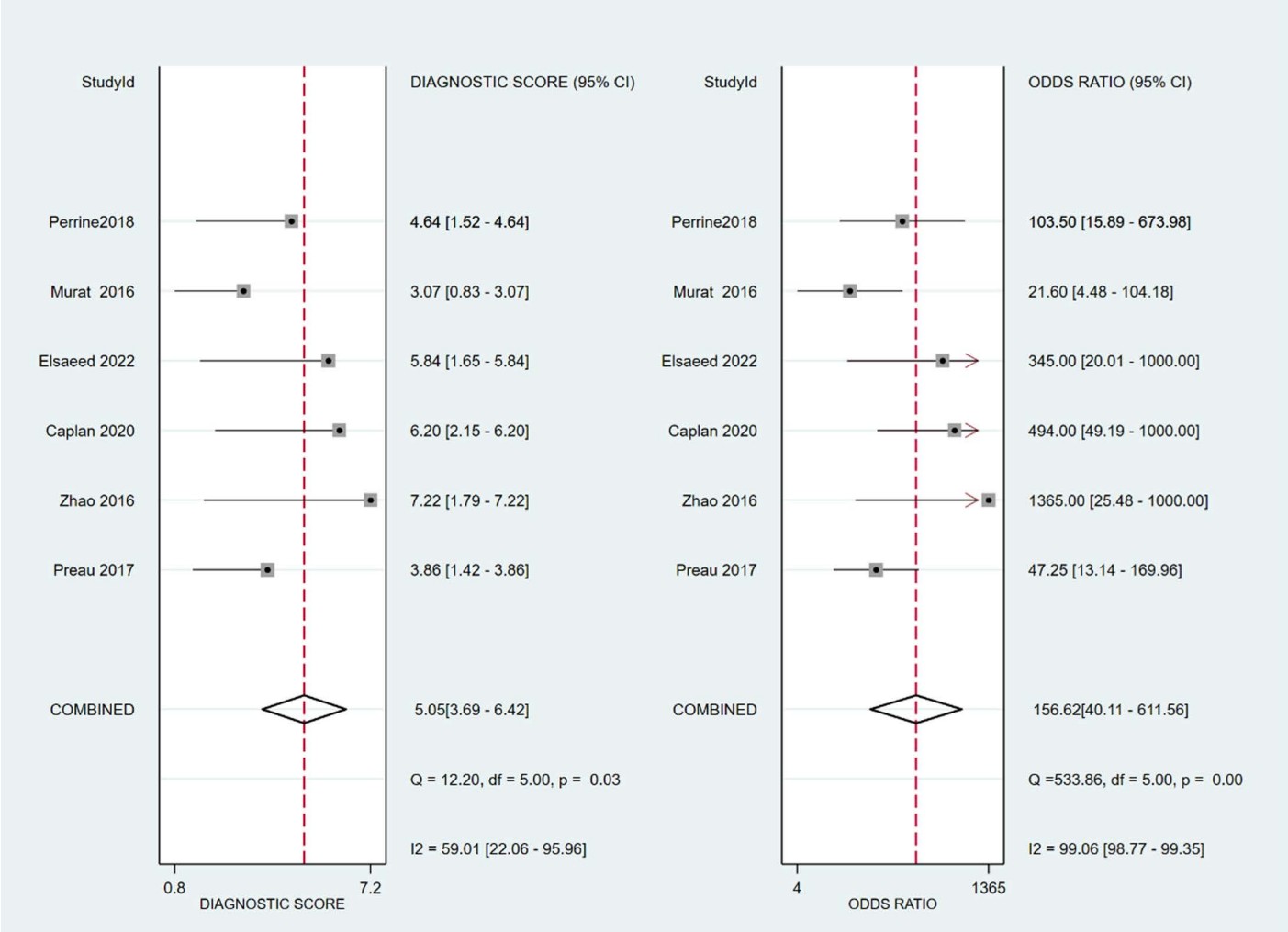

**Fig 15. Forest plot of DOR in the diagnosis of cIVC.**

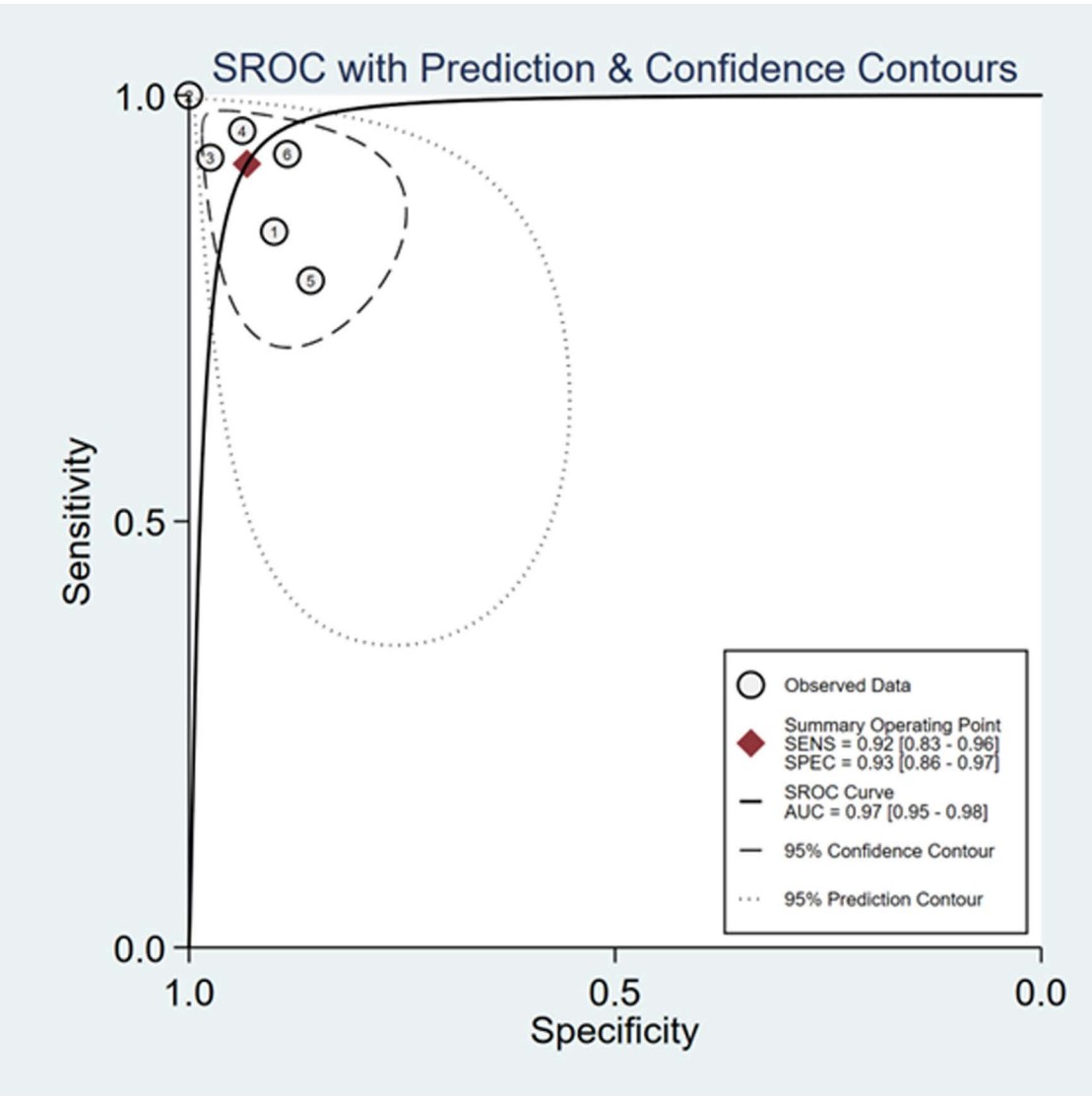

**Fig 16. Summary receiver operator characteristic (SROC) curve in the prediction of cIVC.** SENS = sensitivity, SPEC = specificity, AUC = area under the receiver operating characteristic curve.

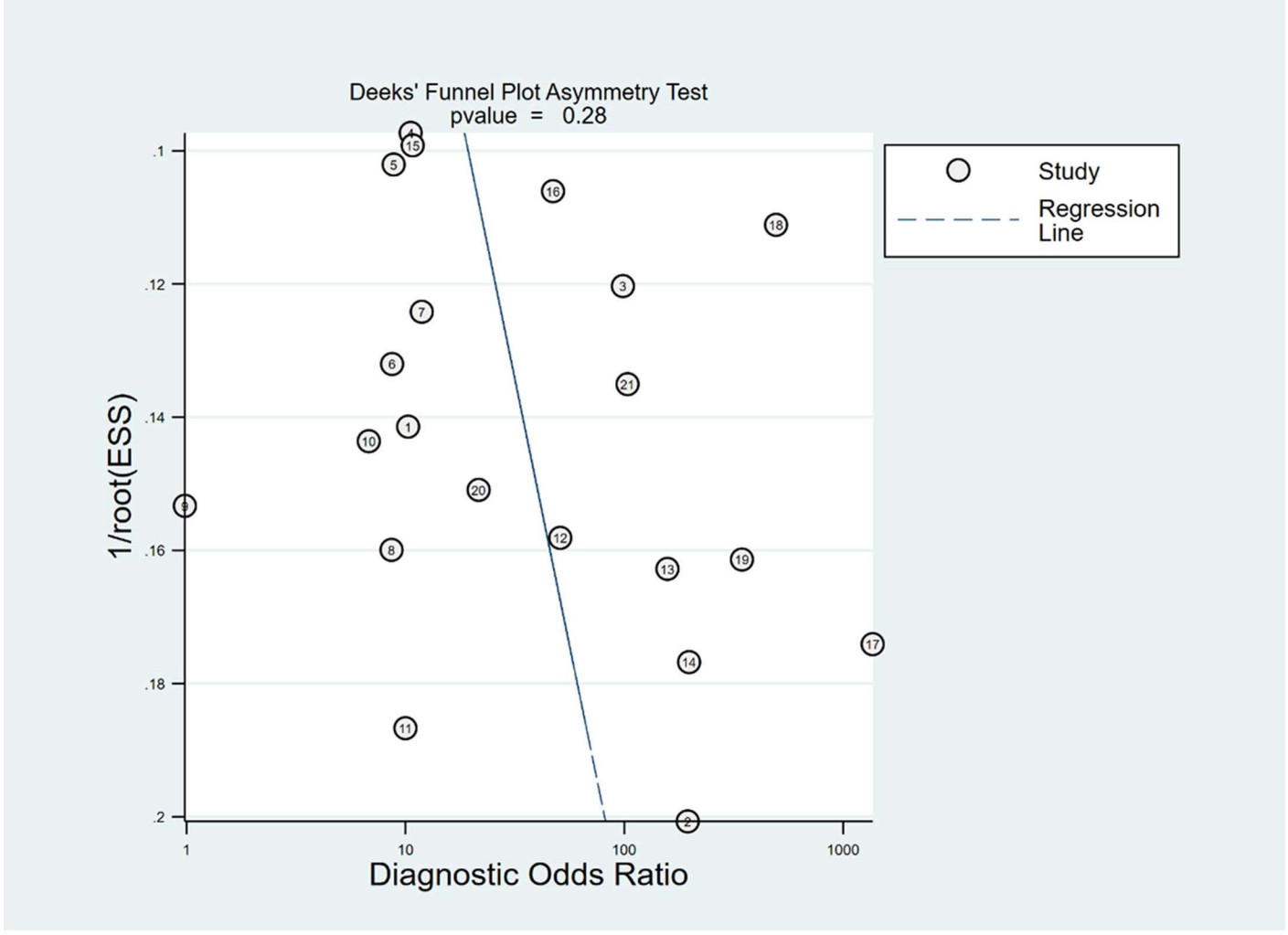

**Fig 17. Deek's funnel plots of ΔIVC** .

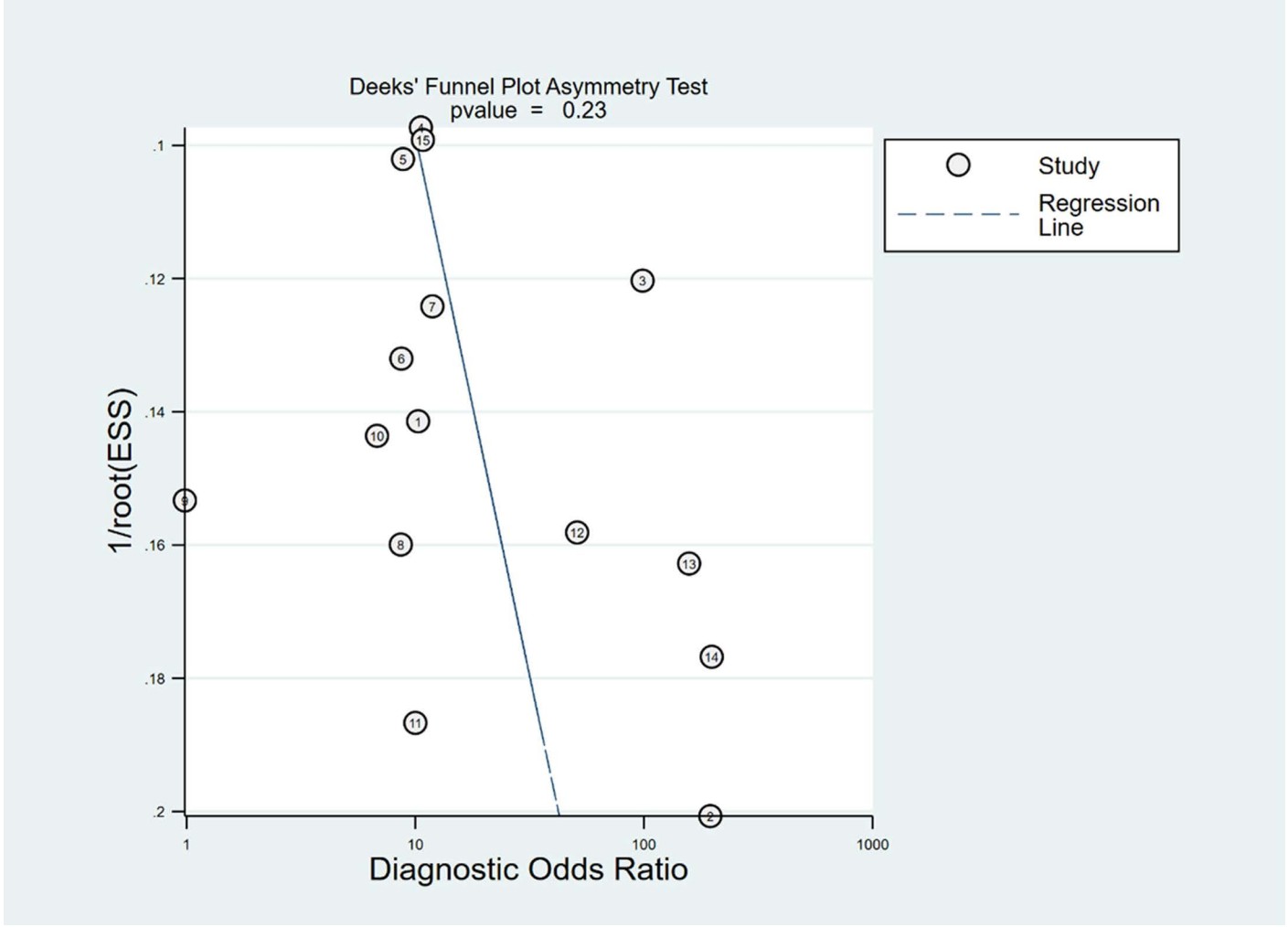

**Fig 18. Deek's funnel plots of dIVC.**

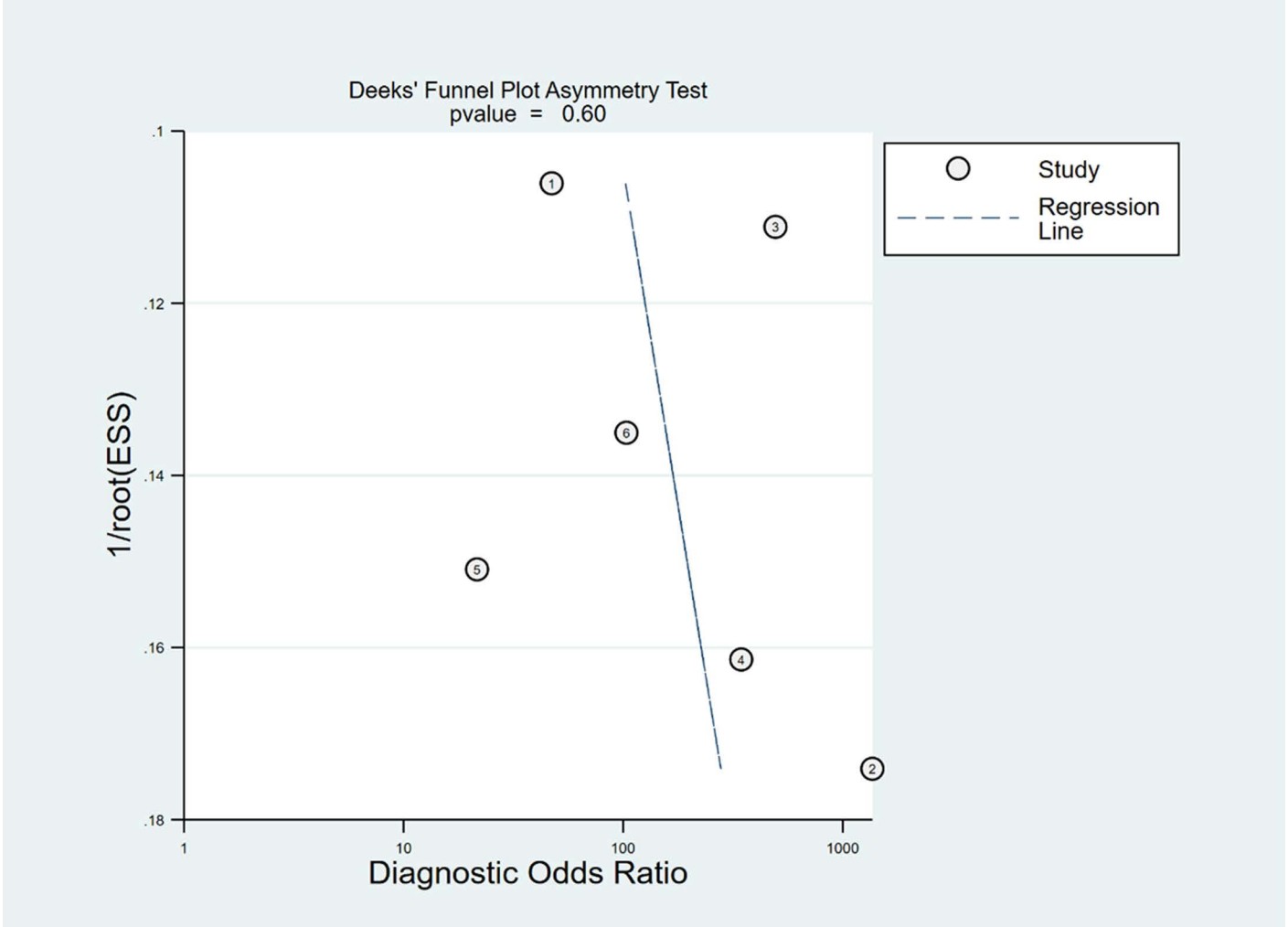

**Fig 19. Deek's funnel plots of cIVC.**

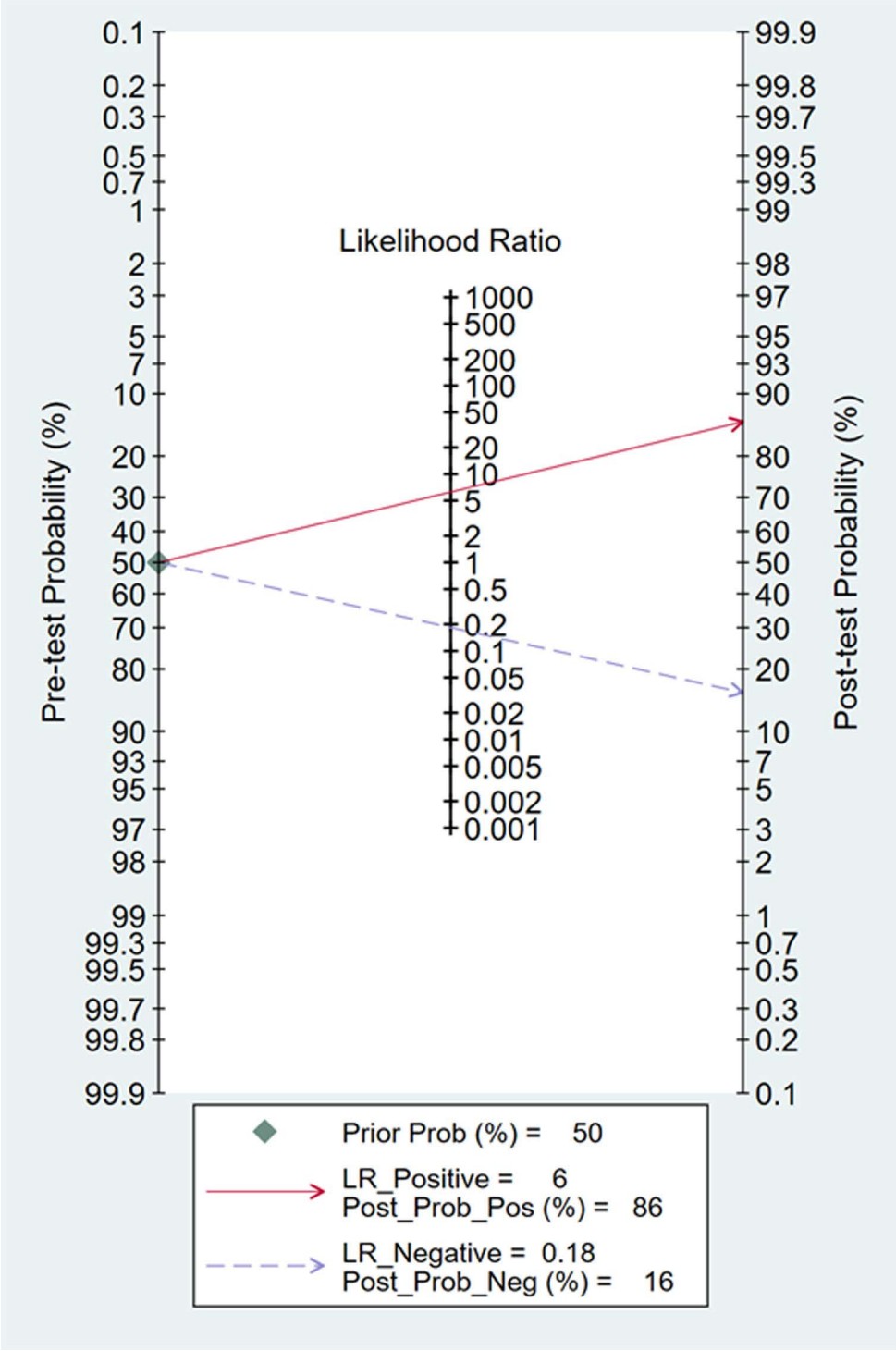

**Fig 20. Fangan plots of ΔIVC.**

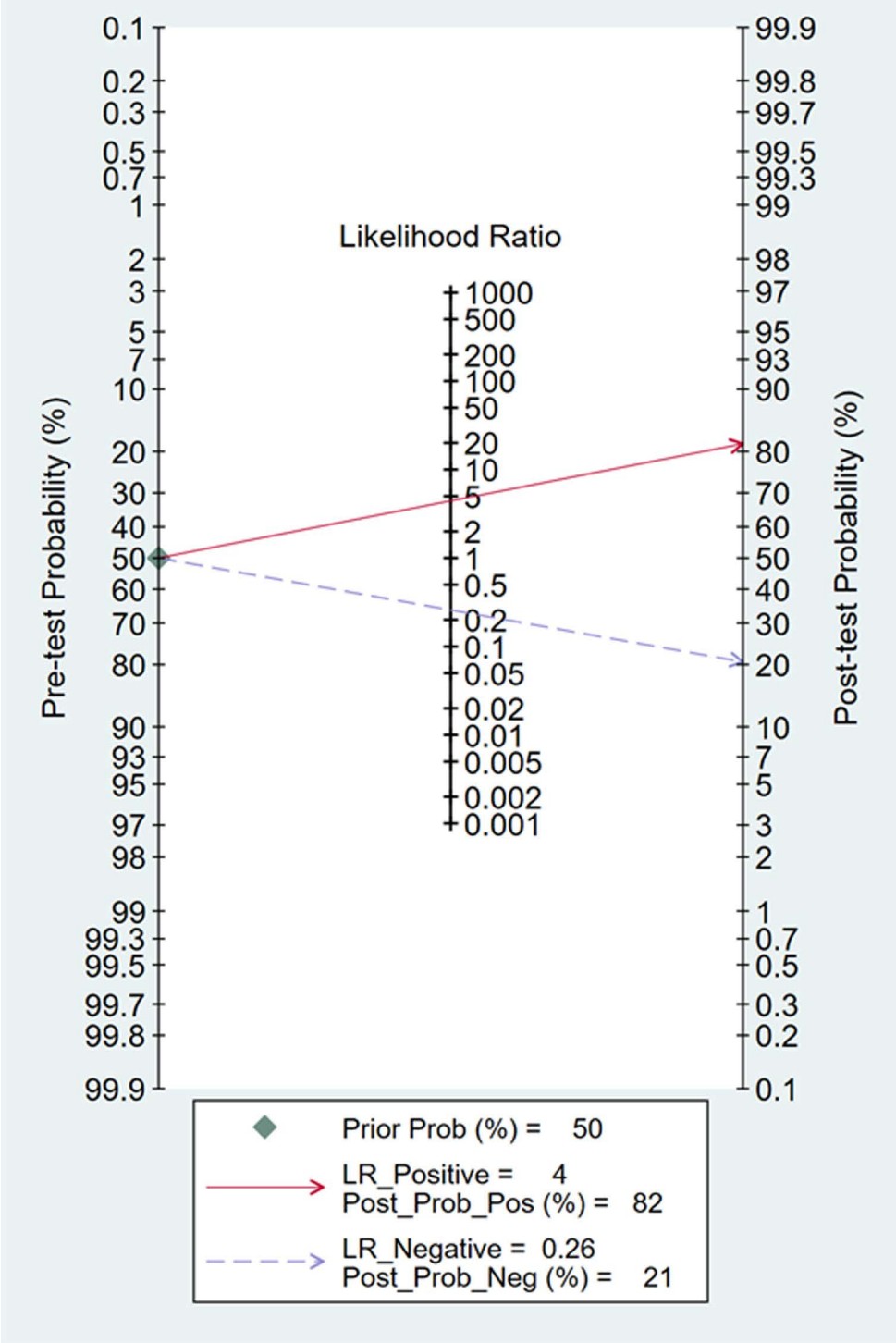

**Fig 21. Fangan plots of dIVC.**

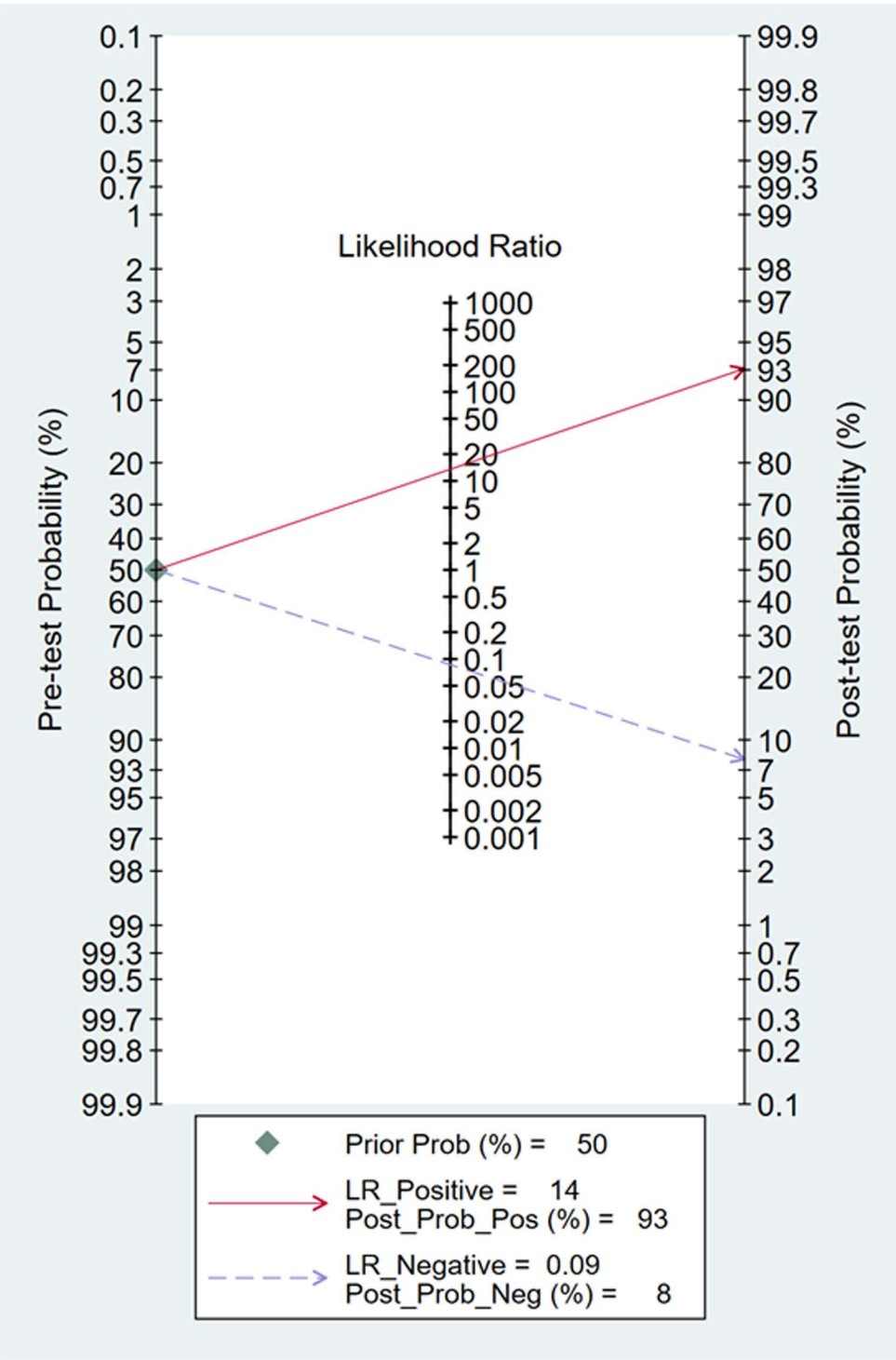

**Fig 22. Fangan plots of cIVC. Subgroup analysis.**

**Table 4. dIVC of Subgroup analyses.**

| Subgroup | sensitivity | specificity | PLR | NLR | DOR | AUC | Meta-regression (*P* value) |
|---|---|---|---|---|---|---|---|
| VT(ml/kg) | | | | | | | 0.580 |
| ≥8 | 0.75 | 0.82 | 3.92 | 0.28 | 16.30 | 0.885 | |
| | 0.68~0.82 | 0.75~0.88 | 1.81~8.50 | 0.13~0.58 | 4~66.49 | 0.816 | |
| ≤8 | 0.85 | 0.85 | 5.61 | 0.20 | 30.96 | 0.888 | |
| | 0.74~0.92 | 0.74~0.92 | 1.58~19.85 | 0.11~0.36 | 5.52~173.68 | 0.819 | |
| PEEP(cmH$_2$O) | | | | | | | 0.732 |
| PEEP≤5 | 0.76 | 0.85 | 4.49 | 0.26 | 18.67 | 0.871 | |
| | 0.67~0.83 | 0.77~0.92 | 2.36~8.54 | 0.14~0.47 | 6.36~54.81 | 0.802 | |
| PEEP≥5 | 0.78 | 0.79 | 3.07 | 0.30 | 10.79 | 0.819 | |
| | 0.67~0.86 | 0.67~0.89 | 1.90~4.99 | 0.18~0.50 | 4.10~28.38 | 0.753 | |
| Infusion volume | | | | | | | 0.417 |
| 500ml | 0.780 | 0.870 | 5.130 | 0.240 | 23.230 | 0.895 | |
| | 0.71~0.84 | 0.79~0.92 | 2.75~9.57 | 0.13~0.45 | 8.84~61.7 | 0.826 | |
| 200ml | 0.64 | 0.85 | 4.02 | 0.42 | 9.78 | 0.753 | |
| | 0.55~0.73 | 0.76~0.91 | 2.52~6.43 | 0.33~0.55 | 5.05~18.97 | 0.696 | |
| ≥7ml/kg | 0.740 | 0.770 | 3.190 | 0.220 | 16.780 | 0.806 | |
| | 0.63~0.83 | 0.65~0.87 | 1.12~9.08 | 0.04~1.08 | 1.51~186.11 | 0.741 | |
| Threshold (%) | | | | | | | 0.472 |
| ≤16.5 | 0.79 | 0.75 | 4.25 | 0.27 | 14.95 | 0.856 | |
| | 0.73~0.83 | 0.69~0.81 | 1.98~9.11 | 0.19~0.40 | 8.14~27.47 | 0.787 | |
| > 16.5 | 0.69 | 0.83 | 3.53 | 0.40 | 10.81 | 0.872 | |
| | 0.62~0.75 | 0.76~0.88 | 1.87~6.64 | 0.24~0.66 | 3.51~33.33 | 0.802 | |

positive likelihood ratio=PLR, negative likelihood ratio=NLR, diagnostic odds ratio=DOR, area under the receiver operating characteristic curve=AUC.

**Table 5. cIVC of Subgroup analyses.**

| Subgroup | sensitivity | specificity | PLR | NLR | DOR | AUC | Meta-regression (*P* value) |
|---|---|---|---|---|---|---|---|
| Infusion volume | | | | | | | 0.791 |
| 500ml | 0.89 | 0.93 | 9.61 | 0.14 | 93.09 | 0.974 | |
| | 0.83~0.94 | 0.86~0.97 | 4.36~21.21 | 0.06~0.29 | 19.52~443.89 | 0.926 | |
| 7ml/kg | 0.958 | 0.937 | / | / | / | 0.970 | |
| | / | / | / | / | / | 0.861~0.999 | |
| PLR | 0.93 | 0.88 | / | / | / | 0.930 | |
| | 0.77~0.99 | 0.69~0.97 | / | / | / | 0.86~0.10 | |
| Threshold (%) | | | | | | | 0.846 |
| > 35 | 0.89 | 0.92 | 10.07 | 0.12 | 101.73 | 0.960 | |
| | 0.82~0.94 | 0.86~0.97 | 4.87~20.80 | 0.06~0.23 | 28.81~359.19 | 0.906 | |
| ≤35 | 0.92 | 0.91 | 7.91 | 0.08 | 138.52 | 0.960 | |
| | 0.84~0.97 | 0.80~0.978 | 3.28~19.09 | 0.01~0.49 | 10.72~1789.77 | 0.903 | |

positive likelihood ratio=PLR, negative likelihood ratio=NLR, diagnostic odds ratio=DOR, area under the receiver operating characteristic curve=AUC.

## Supporting information

**S1. Checklist Table for the present systematic review** .
(DOCX)

**S2. PRISMA_2020_checklist for the present systematic review.**
(DOCX)

**S3. Supplementary information-1.xlsx.**
(XLSX)

## Acknowledgments

We thank all authors of the included studies to help us get the data required in this meta-analysis.

## Author contributions

**Conceptualization:** Hao Zhang, Jingyuan Jiang, Ningxiang Li, Yongli Gao.

**Data curation:** Hao Zhang, Jingyuan Jiang, Yan Liang, Ningxiang Li, Yongli Gao.

**Formal analysis:** Hao Zhang.

**Methodology:** Min Dai, Yan Liang.

**Resources:** Ningxiang Li.

**Software:** Min Dai, Ningxiang Li.

**Writing – original draft:** Hao Zhang, Jingyuan Jiang, Yongli Gao.

**Writing – review & editing:** Hao Zhang, Jingyuan Jiang, Yongli Gao.

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
