## [Decision Letter · Decision Letter 0]

10 Oct 2024

PONE-D-24-36864Accuracy of the inferior vena cava in predicting fluid responsiveness in patients with sepsis: a systematic review and meta-analysisPLOS ONE

Dear Dr. Zhang,

Thank you for submitting your manuscript to PLOS ONE. After careful consideration, we feel that it has merit but does not fully meet PLOS ONE’s publication criteria as it currently stands. Therefore, we invite you to submit a revised version of the manuscript that addresses the points raised during the review process.

**ACADEMIC EDITOR: **
**All issues of both reviewers are required. Please improve english grammar and style of the draft.**

We look forward to receiving your revised manuscript.

Kind regards,

Vincenzo Lionetti, M.D., PhD

Academic Editor

PLOS ONE

**Journal Requirements:**

4. As required by our policy on Data Availability, please ensure your manuscript or supplementary information includes the following: 

Reviewers' comments:

Reviewer's Responses to Questions

**Comments to the Author**

1. Is the manuscript technically sound, and do the data support the conclusions?

Reviewer #1: Yes

Reviewer #2: Partly

2. Has the statistical analysis been performed appropriately and rigorously? 

Reviewer #1: Yes

Reviewer #2: Yes

3. Have the authors made all data underlying the findings in their manuscript fully available?

Reviewer #1: Yes

Reviewer #2: Yes

4. Is the manuscript presented in an intelligible fashion and written in standard English?

Reviewer #1: Yes

Reviewer #2: Yes

5. Review Comments to the Author

**Reviewer #1: ** Dear authors

Appreciated for this effort of reviewing literature related to IVC diameter changes and its relation to fluid responsiveness prediction.

Your efforts are well done.

I just suggest to give more elaboration on the different observation of outcome effect between dIVC and cIVC from physiological point.

Additionally did you ascertain that vasopressors were not manipulated during fluid challenge in included studies as this might affect the standard reference of CI or CO.

Thanks

**Reviewer #2:**  Revisions to improve usefulness:

1. Provide a more detailed discussion of the clinical implications of the different ΔIVC, dIVC, and cIVC thresholds used across studies. What guidance can be given to clinicians on choosing appropriate cutoffs?

2. Expand the analysis and discussion of potential sources of bias in the included studies, particularly related to patient selection and index test conduct.

3. Include a more thorough explanation of the observed heterogeneity between studies and its implications for interpreting the pooled results.

4. Provide clearer guidance on how clinicians should integrate these IVC measurements with other clinical parameters when assessing fluid responsiveness.

5. Discuss any limitations in the generalizability of these findings to different sepsis patient populations or clinical settings.

6. Consider including a proposed algorithm or decision tree for using IVC measurements in clinical practice based on the synthesis of evidence from this review.

Introduction:

The introduction provides a good overview of the clinical importance of fluid management in sepsis and the potential role of IVC measurements. However, it could be strengthened by:

・Providing more context on current guidelines for fluid management in sepsis and their limitations.

・Clarifying the specific advantages of IVC measurements over other methods of assessing fluid responsiveness.

Methods:

The methods section is generally well-described, but could be improved by:

・Providing more detail on the search strategy, including full search terms for each database.

・Clarifying the criteria used for assessing study quality and how this information was incorporated into the analysis.

・Explaining how discrepancies in data extraction or quality assessment between reviewers were resolved.

Results:

The results are comprehensively presented, but some aspects could be clarified:

・The rationale for the chosen subgroup analyses could be more explicitly stated.

・More detail on the results of the quality assessment for individual studies would be helpful.

・A more in-depth exploration of the sources of heterogeneity would strengthen the analysis.

Discussion:

The discussion provides a good overview of the findings, but could be enhanced by:

・A more critical analysis of the limitations of the included studies and how these might impact the overall conclusions.

・More detailed discussion of the clinical implications of the different thresholds used across studies.

・Clearer guidance on how clinicians should integrate these IVC measurements with other clinical parameters when assessing fluid responsiveness.

・A more thorough comparison of these findings with other methods of assessing fluid responsiveness in sepsis.

6. PLOS authors have the option to publish the peer review history of their article (what does this mean? ). If published, this will include your full peer review and any attached files.

**Do you want your identity to be public for this peer review?** For information about this choice, including consent withdrawal, please see our Privacy Policy .

Reviewer #1: **Yes: ** Fadi Aljamaan

Reviewer #2: No

---

## [Author Response · Author response to Decision Letter 1]

27 Nov 2024

Dear Reviewers,

Happy Thanksgiving Day! Thank you for your thorough review and constructive suggestions regarding our manuscript. We have revised the manuscript in accordance with your comments and have marked all amendments in the revised document. In addition to our point-by-point responses to the reviewers’ comments, we have also made every effort to meet the required editorial standards, ensuring that all changes are easily identifiable. We hope that our revised manuscript fulfills your requirements. If any further action is needed, please do not hesitate to let us know. We look forward to your feedback.

1. Please improve english grammar and style of the draft

Response: Considering the academic editor’s suggestion, we have engaged a native English speaker to thoroughly revise our language, ensuring the fluency, accuracy, and consistency of our writing. We believe these changes will greatly enhance the quality of our manuscript.

2. Data Availability Statement

Response: Considering the Reviewer’s suggestion, we have corrected it, and it has been changed to “All data are in the manuscript and/or supporting information files”. Please see the Data Availability Statement in the attachment of Supplementary Material.

3. Literature search section

Response: Considering the Reviewer’s suggestion, we uploaded the files in Excel format of Supplementary information.

4. Suggest to Give more elaboration on the different observation of outcome effect between dIVC and cIVC from physiological point.

Response: Considering the Reviewer’s suggestion, we modified the Introduction and Discussion accordingly and added the following references. As detailed in the Introduction, page 6, lines 100 -103, see Discussion section, page 14, line 278-287.

5. The reviewer raises an important point that vasopressors were not manipulated during fluid challenge in included studies as this might affect the standard reference of CI or CO?

Response: Considering the Reviewer’s suggestion, all members of the research team read all the literatures. We reviewed the full text and found that four literatures did not explicitly mention the use of vasopressors, two literatures excluded patients who were using vasopressors, and the remaining literatures all acknowledged the use of vasopressors during the study period. Subgroup analysis was not feasible due to the limited number of relevant studies. However, this issue is addressed in the Discussion and Limitations sections. Moreover, the details of the literature have been incorporated into Table1 Main characteristics of the eligible studies by the author.

6. Providing more context on current guidelines for fluid management in sepsis and their limitations.

Response: Considering the Reviewer’s suggestion, we modified the Introduction accordingly. As detailed in the Introduction, page 4, lines 57 -64.

7. Clarifying the specific advantages of IVC measurements over other methods of assessing fluid responsiveness.

Response: Considering the Reviewer’s suggestion, we modified the Introduction accordingly. As detailed in the Introduction, page 5, lines 74-78, 82-85.

8. Providing more detail on the search strategy, including full search terms for each database.

Response: Thank you for the comment. We provided a complete search strategy for the PubMed database which showed in Fig 1. We now have elaborated the search strategy in Word format giving the details of Electronic search strategy.

9. Clarifying the criteria used for assessing study quality and how this information was incorporated into the analysis.

Response: Considering the Reviewer’s suggestion, we used the QUADAS 2 tool as a guide to direct our judgments as to the quality of the included studies. ZH scored and recorded for all references using the QUADAS-2 tool, and then DM reviewed the results. More detailed descriptions in the supplementary Excel file were also provided.

10. Explaining how discrepancies in data extraction or quality assessment between reviewers were resolved.

Response: Considering the Reviewer’s suggestion, we modified the Materials and Methods accordingly. In case of disagreement between two researchers in the literature screening or data extraction process, the decision was submitted to the third researcher (GYL). As detailed in the Study selection, page 8, lines 142-144.

11. The rationale for the chosen subgroup analyses could be more explicitly stated.

Response: Considering the Reviewer’s suggestion, we modified the Subgroup analysis accordingly. As detailed in the Subgroup analysis, page 11, lines 210-213.

12. More detail on the results of the quality assessment for individual studies would be helpful.

Response: Considering the Reviewer’s suggestion, we used the QUADAS 2 tool as a guide to direct our judgments as to the quality of the included studies. More detailed descriptions in the supplementary Excel file were also provided.

13. A more in-depth exploration of the sources of heterogeneity would strengthen the analysis.

Response: Considering the Reviewer’s suggestion, we modified the Discussion accordingly. see Discussion section, page 12,14-16.

14. More detailed discussion of the clinical implications of the different thresholds used across studies.

Response: Considering the Reviewer’s suggestion, we modified the Discussion accordingly. see Discussion section, page 16, lines 333-340.

15. Clearer guidance on how clinicians should integrate these IVC measurements with other clinical parameters when assessing fluid responsiveness.

Response: Considering the Reviewer’s suggestion, we modified the Discussion accordingly. see Discussion section, page 14, lines 279-288.

16. A more thorough comparison of these findings with other methods of assessing fluid responsiveness in sepsis.

Response: Considering the Reviewer’s suggestion, we modified the Discussion accordingly. see Discussion section, page 12,14-16.

---

## [Decision Letter · Decision Letter 1]

17 Dec 2024

PONE-D-24-36864R1Predictive accuracy of changes in the inferior vena cava diameter for predicting fluid responsiveness in patients with sepsis: A systematic review and meta-analysisPLOS ONE

Dear Dr. Zhang,

Thank you for submitting your manuscript to PLOS ONE. After careful consideration, we feel that it has merit but does not fully meet PLOS ONE’s publication criteria as it currently stands. Therefore, we invite you to submit a revised version of the manuscript that addresses the points raised during the review process.

ACADEMIC EDITOR: Despite some improvement, the revised form requires further clarification in accord with Reviewer's suggestions/>==============================

We look forward to receiving your revised manuscript.

Kind regards,

Vincenzo Lionetti, M.D., PhD

Academic Editor

PLOS ONE

Reviewers' comments:

Reviewer's Responses to Questions

Comments to the Author

1. If the authors have adequately addressed your comments raised in a previous round of review and you feel that this manuscript is now acceptable for publication, you may indicate that here to bypass the “Comments to the Author” section, enter your conflict of interest statement in the “Confidential to Editor” section, and submit your "Accept" recommendation.

Reviewer #1: All comments have been addressed

Reviewer #2: (No Response)

2. Is the manuscript technically sound, and do the data support the conclusions?

Reviewer #1: Yes

Reviewer #2: Yes

3. Has the statistical analysis been performed appropriately and rigorously? 

Reviewer #1: Yes

Reviewer #2: No

4. Have the authors made all data underlying the findings in their manuscript fully available?

Reviewer #1: Yes

Reviewer #2: No

5. Is the manuscript presented in an intelligible fashion and written in standard English?

Reviewer #1: Yes

Reviewer #2: No

6. Review Comments to the Author

Reviewer #1: Appreciated for the excellent work.

It is still as you mentioned in the limitations that pattern and type of breathing affect the change of dIVC and its predictivity even though you could not find statistical significance with PEEP and VT, but for sure many other non recorded variable may play role.

Reviewer #2: (No Response)

7. PLOS authors have the option to publish the peer review history of their article (what does this mean? ). If published, this will include your full peer review and any attached files.

Do you want your identity to be public for this peer review? For information about this choice, including consent withdrawal, please see our Privacy Policy .

Reviewer #1: Yes: Aljamaan, Fadi S

Reviewer #2: No

---

## [Author Response · Author response to Decision Letter 2]

29 Dec 2024

Dear Reviewers,

Thank you for your thorough review and constructive suggestions regarding our manuscript. We have revised the manuscript in accordance with your comments and have marked all amendments in the revised document. In addition to our point-by-point responses to the reviewers’ comments, we have also made every effort to meet the required editorial standards, ensuring that all changes are easily identifiable. We hope that our revised manuscript fulfills your requirements. If any further action is needed, please do not hesitate to let us know. We look forward to your feedback.

1. It is still as you mentioned in the limitations that pattern and type of breathing affect the change of dIVC and its predictivity even though you could not find statistical significance with PEEP and VT, but for sure many other non recorded variable may play role.

Response Reviewer #1: Considering the Reviewer’s suggestion, we have expounded more detail in the discussion section, page 13, line 252-255. Based on past research findings, we suggest that this outcome could be attributed to physiological differences between adults and children. They included several studies of pediatric patients, whereas we excluded these studies.

2. Is the manuscript presented in an intelligible fashion and written in standard English?

Response Reviewer #2: Considering the Reviewer’s suggestion, we have engaged a native English speaker to thoroughly revise our language, ensuring the fluency, accuracy, and consistency of our writing. We believe these changes will greatly enhance the quality of our manuscript.

3. Has the statistical analysis been performed appropriately and rigorously?

Response Reviewer #2: This systematic review and meta-analysis was carried out was conducted in accordance with the Preferred Reporting Items for Systematic Reviews and Meta-Analysis of Diagnostic Test Accuracy (PRISMA-DTA) statement. Based on the given guidelines, we employed the statistical analysis software provided by the instructions to obtain the data results.

4. Have the authors made all data underlying the findings in their manuscript fully available?

Response Reviewer #2: Considering the Reviewer’s suggestion, All of our data, which is open and transparent, has been uploaded to the attachment in Excel tables.

---

## [Decision Letter · Decision Letter 2]

7 Jan 2025

PONE-D-24-36864R2Predictive accuracy of changes in the inferior vena cava diameter for predicting fluid responsiveness in patients with sepsis: A systematic review and meta-analysisPLOS ONE

Dear Dr. Zhang,

Thank you for submitting your manuscript to PLOS ONE. After careful consideration, we feel that it has merit but does not fully meet PLOS ONE’s publication criteria as it currently stands. Therefore, we invite you to submit a revised version of the manuscript that addresses the points raised during the review process.

**ACADEMIC EDITOR: **

The authors are encouraged to reply to all comments by one reviewer.

We look forward to receiving your revised manuscript.

Kind regards,

Vincenzo Lionetti, M.D., PhD

Academic Editor

PLOS ONE

Reviewers' comments:

Reviewer's Responses to Questions

**Comments to the Author**

1. If the authors have adequately addressed your comments raised in a previous round of review and you feel that this manuscript is now acceptable for publication, you may indicate that here to bypass the “Comments to the Author” section, enter your conflict of interest statement in the “Confidential to Editor” section, and submit your "Accept" recommendation.

Reviewer #2: (No Response)

2. Is the manuscript technically sound, and do the data support the conclusions?

Reviewer #2: Yes

3. Has the statistical analysis been performed appropriately and rigorously? 

Reviewer #2: Yes

4. Have the authors made all data underlying the findings in their manuscript fully available?

Reviewer #2: No

5. Is the manuscript presented in an intelligible fashion and written in standard English?

Reviewer #2: Yes

6. Review Comments to the Author

Reviewer #2: This systematic review and meta-analysis examines the predictive accuracy of inferior vena cava (IVC) measurements for fluid responsiveness in septic patients. The study analyzed 21 papers (1,207 patients), evaluating three key parameters: overall IVC diameter changes (∆IVC), IVC distensibility (dIVC) in mechanically ventilated patients, and IVC collapsibility (cIVC) in spontaneously breathing patients. The results demonstrated good diagnostic accuracy across all measures, with particularly strong performance of cIVC in spontaneously breathing patients. However, there are some concerns needing revision.

Methodology:

1. The selection criteria differentiating mechanically ventilated from spontaneously breathing patients needs clearer exposition

2. The heterogeneity in ventilator parameters across studies requires more detailed analysis

3. The rationale for subgroup analysis thresholds should be better justified

Results Presentation:

1. Forest plots and funnel plots should be integrated into the main text rather than supplementary materials for better readability

2. The discussion of varying thresholds across studies needs expansion

3. The implications of different fluid challenge protocols require more detailed analysis

4. The meta-regression results need clearer presentation, particularly regarding the clinical significance of findings

Discussion:

1. The limitations section should be expanded to address:

- Heterogeneity in study protocols

- Variations in measurement techniques

- Impact of different patient populations

2. Clinical implementation challenges need more thorough discussion

3. Specific recommendations for standardizing measurement techniques should be included

4. The implications of different ventilator settings need more detailed analysis

7. PLOS authors have the option to publish the peer review history of their article (what does this mean? ). If published, this will include your full peer review and any attached files.

**Do you want your identity to be public for this peer review?** For information about this choice, including consent withdrawal, please see our Privacy Policy .

Reviewer #2: No

---

## [Author Response · Author response to Decision Letter 3]

23 Feb 2025

Dear Editors,

Thank you for your thorough review and constructive suggestions regarding our manuscript. We sincerely appreciate the time and effort you dedicated to reviewing and providing feedback. In particular, we highly value the constructive suggestions you made, which were carefully discussed within our team. We fully agree with your recommendations and have made detailed revisions accordingly. These improvements will help enhance the manuscript’s quality and ensure it meets the standards for publication. Once again, thank you for your valuable input, which has greatly contributed to this research. We look forward to your feedback. Below are our point-by-point responses to your comments:

Methodology:

1. The selection criteria differentiating mechanically ventilated from spontaneously breathing patients needs clearer exposition.

Response academic editor: Thank you once again for your constructive feedback. We greatly appreciate and value your insights. In response, we have revised and enhanced the Methodology section. Your valuable input has improved the completeness of the description.

The manuscript has been amended to state: “The inclusion criteria included studies published in Chinese and English that focused on patients with sepsis or septic shock. The research subjects in the original literature should identify whether they are mechanically ventilated or spontaneously breathing patients. Mechanically ventilated patients should collect ventilator parameters.”

2. The heterogeneity in ventilator parameters across studies requires more detailed analysis

Response academic editor: Thank you for your valuable suggestion. In response, we have revised the Methodology section accordingly. And it has been changed to “For the subgroup analysis, we first compared mechanically ventilated and nonmechanically ventilated individuals. We analyzed various ventilator parameters including TV, PEEP, and threshold, which are mechanically ventilated.”

3. The rationale for subgroup analysis thresholds should be better justified

Response academic editor: Thank you for your helpful suggestion. In response, we have revised the Methodology section accordingly. The manuscript has been amended to state: “Currently, there is no evidence that a single evaluation parameter or index can be utilized as an endpoint on its own to direct fluid resuscitation in sepsis patients. Our subgroup analysis is based on previous systematic review results, basic ventilator parameters (TV=6-8mL/kg, PEEP=5 cmH2O), and dose selection for fluid resuscitation. Each enrolled patient with septic shock was categorized into the low-volume (200ml), medium-volume (7 mL/kg fluid), or high-volume (above 500 mL) fluid group according to the infusion volume. We conducted subgroup analyses based on TV, PEEP, infusion volume, and diagnostic threshold.”

Results Presentation:

1. Forest plots and funnel plots should be integrated into the main text rather than supplementary materials for better readability

Response academic editor: Thank you for your insightful comment and helpful suggestions. In response, we have revised the forest plots and funnel plots accordingly.

2. The discussion of varying thresholds across studies needs expansion?

Response academic editor: Thank you for your insightful comment and helpful suggestions. In response, we have revised the Results and Discussion section accordingly.

3. The implications of different fluid challenge protocols require more detailed analysis.

Response academic editor: Thank you for your insightful suggestion. In response, we have revised the Results and Discussion section accordingly.

4. The meta-regression results need clearer presentation, particularly regarding the clinical significance of findings.

Response academic editor: Thank you for your helpful suggestion. In response, we have revised the Results and Discussion section accordingly.

Discussion Presentation:

1. The limitations section should be expanded to address: Heterogeneity in study protocols, Variations in measurement techniques, Impact of different patient populations.

Response academic editor: Thank you for your helpful suggestion. In response, we have revised the Limitations section accordingly.

“First, heterogeneity in measurement protocols primarily arises from differences between mechanically ventilated and non-ventilated patient cohorts. Second, critically ill patients demonstrate significant heterogeneity that those with heart failure, kidney disease, or other comorbidities may exhibit marked variations in fluid resuscitation efficacy. This necessitates the integration of dynamic parameters, such as ultrasound-guided assessments combined with pulse pressure variation (PPV) or fluid challenge tests, when evaluating fluid resuscitation in septic patients. Additionally, blood lactate levels should be combined to comprehensively assess resuscitation outcomes.”

2. Clinical implementation challenges need more thorough discussion.

Response academic editor: Thank you for your thoughtful suggestion. In response, we have revised the Discussion section accordingly.

3. Specific recommendations for standardizing measurement techniques should be included.

Response academic editor: Thank you for your suggestion. In response, we have revised the Discussion section accordingly.

4. The implications of different ventilator settings need more detailed analysis.

Response academic editor: Thank you for your suggestion. In response, we have revised the Discussion section accordingly.

---

## [Decision Letter · Decision Letter 3]

17 Mar 2025

Predictive accuracy of changes in the inferior vena cava diameter for predicting fluid responsiveness in patients with sepsis: A systematic review and meta-analysis

PONE-D-24-36864R3

Dear Dr. Zhang,

We’re pleased to inform you that your manuscript has been judged scientifically suitable for publication and will be formally accepted for publication once it meets all outstanding technical requirements.

Kind regards,

Vincenzo Lionetti, M.D., PhD

Academic Editor

PLOS ONE

Additional Editor Comments (optional):

Reviewers' comments:

Reviewer's Responses to Questions

**Comments to the Author**

1. If the authors have adequately addressed your comments raised in a previous round of review and you feel that this manuscript is now acceptable for publication, you may indicate that here to bypass the “Comments to the Author” section, enter your conflict of interest statement in the “Confidential to Editor” section, and submit your "Accept" recommendation.

Reviewer #2: All comments have been addressed

2. Is the manuscript technically sound, and do the data support the conclusions?

Reviewer #2: Yes

3. Has the statistical analysis been performed appropriately and rigorously? 

Reviewer #2: Yes

4. Have the authors made all data underlying the findings in their manuscript fully available?

Reviewer #2: Yes

5. Is the manuscript presented in an intelligible fashion and written in standard English?

Reviewer #2: Yes

6. Review Comments to the Author

Reviewer #2: After reviewing the revised manuscript, I find that the authors have addressed the major concerns raised in the previous review. The improvements to the methodology, results presentation, and discussion sections have significantly enhanced the quality and clarity of the manuscript.

7. PLOS authors have the option to publish the peer review history of their article (what does this mean? ). If published, this will include your full peer review and any attached files.

**Do you want your identity to be public for this peer review?** For information about this choice, including consent withdrawal, please see our Privacy Policy .

Reviewer #2: No

---

## [Editor Report · Acceptance letter]

PONE-D-24-36864R3

PLOS ONE

Dear Dr. Zhang,

I'm pleased to inform you that your manuscript has been deemed suitable for publication in PLOS ONE. Congratulations! Your manuscript is now being handed over to our production team.

Kind regards,

on behalf of

Prof. Vincenzo Lionetti

Academic Editor

PLOS ONE